# Systemic delivery of full-length dystrophin in Duchenne muscular dystrophy mice

Yuan Zhou [1,2], Chen Zhang[1], Weidong Xiao [1], Roland W. Herzog [1] & Renzhi Han [1] ✉

Current gene therapy for Duchenne muscular dystrophy (DMD) utilizes adeno-associated virus (AAV) to deliver micro-dystrophin (μDys), which does not provide full protection for striated muscles as it lacks many important functional domains of full-length (FL) dystrophin. Here we develop a triple vector system to deliver FL-dystrophin into skeletal and cardiac muscles. We split FL-dystrophin into three fragments linked to two orthogonal pairs of split intein, allowing efficient assembly of FL-dystrophin. The three fragments packaged in myotropic AAV (MyoAAV4A) restore FL-dystrophin expression in both skeletal and cardiac muscles in male $mdx^{4cv}$ mice. Dystrophin-glycoprotein complex components are also restored at the sarcolemma of dystrophic muscles. MyoAAV4A-delivered FL-dystrophin significantly improves muscle histopathology, contractility, and overall strength comparable to μDys, but unlike μDys, it also restores defective cavin 4 localization and associated signaling in $mdx^{4cv}$ heart. Therefore, our data support the feasibility of a mutation-independent FL-dystrophin gene therapy for DMD, warranting further clinical development.

Duchenne muscular dystrophy (DMD) is a fatal genetic disease afflicting approximately 1 in 3800–6300 live male births and caused by genetic mutations in the *DMD* gene located on the X chromosome[1]. The dystrophin protein encoded by the *DMD* gene belongs to the spectrin superfamily of cytoskeletal proteins[2]. The major isoform expressed in striated muscles is a 427 kDa protein, containing four main domains: an N-terminal actin-binding domain, a long rod-like domain composed of spectrin-like repeats, a cysteine-rich domain that binds to dystroglycan and other membrane-associated proteins, and a C-terminal domain that interacts with syntrophin and α-dystrobrevin, forming a large dystrophin-glycoprotein complex (DGC)[3,4]. The DGC links the actin cytoskeleton to the extracellular matrix, facilitating force transmission, providing mechanical stability and membrane protection, and mediating signal transduction[5–7]. Disrupted expression of dystrophin leads to repeated cycles of muscle damage and repair, which eventually exhaust the regenerative capacity of muscle stem cells and cause fibrosis and fatty replacement. The progressive loss of muscle mass and function in the skeletal, cardiac, and respiratory muscles eventually leads to impaired mobility, cardiomyopathy, respiratory failure, and early death[8,9].

A gene replacement therapy using adeno-associated virus (AAV) to deliver a correct copy of the *DMD* gene could allow the restoration of dystrophin to halt/reverse the deterioration of muscles[10]. However, the limited cargo capacity of AAV (~4.5 kb) makes it challenging to deliver the FL *DMD* gene (cDNA is over 11 kb). Early studies found that patients carrying a large internal deletion of the *DMD* gene developed a milder form of the disease, known as Becker muscular dystrophy (BMD), due to the expression of truncated yet partially functional dystrophin proteins[11]. These findings led to the development of micro-dystrophins (μDys) gene therapy, which aims to restore the expression of a truncated version of the dystrophin protein[12–14]. These miniaturized forms of dystrophin are only about 1/3 of the FL-dystrophin, retain some of the essential functional domains of the protein and can be packaged into one AAV vector for in vivo delivery. This approach has shown promise in preclinical animal models, restoring muscle integrity and improving muscle function[13–15]. Recently, the US Food and Drug

[1]Department of Pediatrics, Herman B Wells Center for Pediatric Research, Indiana University School of Medicine, Indianapolis, IN 46202, USA. [2]Present address: Department of Thoracic Surgery, Xiangya Hospital, Central South University, 410008 Changsha, Hunan, China. ✉e-mail: rh11@iu.edu

Administration (FDA) granted accelerated approval to Sarepta's Elevidys (µDys delivered in AAVrh.74) for DMD patients aged 4 to 5 years[16]. However, due to the lack of 2/3 of the dystrophin coding sequence containing critical rod and hinge domains of dystrophin needed to connect with other proteins in the dystrophin complex, these µDys gene therapies are unable to provide full protection of skeletal and heart muscle integrity and function[15,17–19]. This limitation highlights the urgent need to develop novel strategies to deliver larger or even FL-dystrophin.

Several different approaches have been explored using dual AAV vectors to deliver larger payloads, including fragmented genome assembly, overlapping, trans-splicing, and hybrid approaches. For instance, two vectors containing an overlapping fragment to express large transgenes may undergo homologous recombination[20–23]. Upon co-delivery, the efficiency of homologous recombination is, however, a limiting factor for this strategy. Another approach known as trans-splicing is to take advantage of the natural concatamerization ability of AAV to expand the transgene size using dual AAV vectors[24,25]. However, the head-to-tail concatamer formation and the trans-splicing of the pre-mRNA across the ITR junction pose rate-limiting steps. These approaches may be combined as shown for hybrid vector systems[26]. Despite the varying success of these dual AAV vector strategies in preclinical studies, the efficiency of expression from the available dual-vector systems is insufficient for many clinical gene therapy applications. It is even more challenging to deliver the FL-dystrophin sequence, which is about three times the packaging capacity of AAV, thus requiring at least three vectors to package the entire coding sequence. A previous attempt with triple trans-splicing AAV vectors showed the feasibility of expressing FL-dystrophin, albeit with only very low efficiency[27].

Split inteins are small polypeptides that self-assemble and undergo a protein trans-splicing (PTS) reaction, resulting in the formation of a mature, fully functional protein in a "traceless manner". We and others have successfully delivered the oversized base editors in dual AAV vectors in vivo using the split intein strategy[28–30]. Leveraged on this success, we developed a triple AAV system with orthogonal split inteins to deliver FL-dystrophin protein. Packaged into an engineered myotropic AAV capsid[31] (MyoAAV4A), this triple vector combination restored the expression of FL-dystrophin and significantly improved muscle histopathology and function in a mouse model of DMD.

## Results

### Proof-of-concept design of split intein constructs to assemble FL-dystrophin

The cDNA for FL-dystrophin is over 11 kb, about three times of the AAV packaging capacity. Thus, three AAV vectors are required to package the entire coding sequence of FL-dystrophin. We rationally split FL-dystrophin cDNA into three fragments based on (1) the fragment size, (2) the protein domain structure, and (3) the junctional sequence compatibility for split intein. We previously utilized Cfa[32] and Gp41-1[33], two orthogonal inteins with a remarkably fast rate of protein trans-splicing (PTS), to successfully assemble adenine base editor (ABE)[28]. We, therefore, chose these two orthogonal pairs of inteins for our initial test. The first split site was chosen in between spectrin repeat (SR) 8 and 9, where a cysteine residue is followed by a bulky tryptophan residue as required for efficient protein splicing by the Cfa intein (Fig. 1a). The second split site was selected within the end of Hinge 3 (H3) domain, where two consecutive serine residues are located, which may facilitate protein splicing by the Gp41-1 intein (Fig. 1a). In addition, this is also where the native Dp140 isoform starts. We named the three fragment constructs of this first version as Dys-N1, Dys-M1, and Dys-C1, respectively. All expression cassettes utilized a mini-CMV promoter with a muscle creatine kinase enhancer (meCMV). Transfection of Dys-N1/M1/C1 into HEK293 cells resulted in the expression of FL-dystrophin detectable by three different anti-dystrophin antibodies that

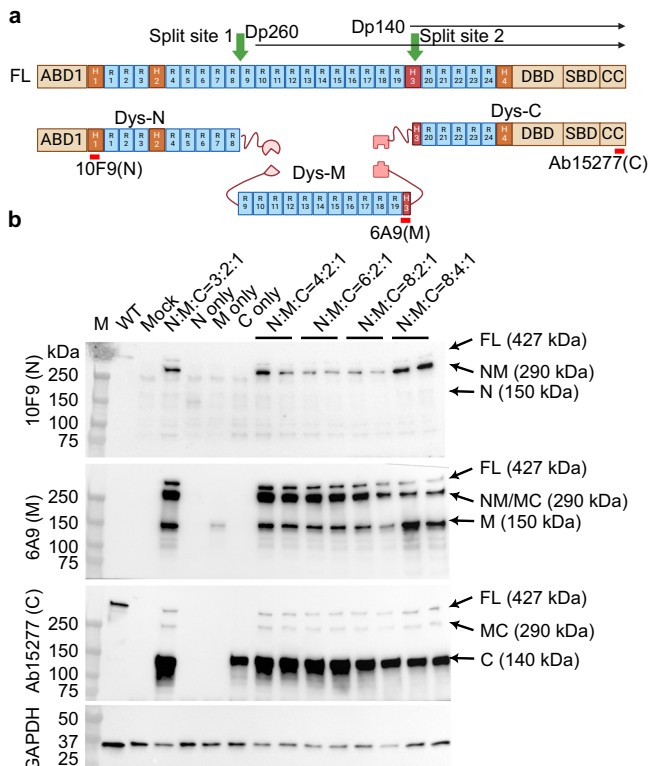

**Fig. 1 | Design and test of split intein constructs to assemble FL-dystrophin. a** Diagram showing the domain structure of FL-dystrophin, the two split sites, the relative initiation sites of Dp260 and Dp140 isoforms, and the three split fragments fused with inteins (Dys-N1, M1, and C1). The antigen epitopes for three different dystrophin antibodies (anti-N, anti-M, and anti-C) were also labeled. **b**, Western blotting analysis of dystrophin expression in HEK293 cells transfected with or without the Dys-N1, M1, and C1 constructs at different molar ratios. WT mouse skeletal muscle lysate was loaded as a positive control and the GAPDH was used as a loading control. This experiment was repeated independently three times with similar results. Source data are provided as a Source Data file.

specifically recognize the N, M, or C fragments, respectively (Fig. 1b). However, the FL-dystrophin band was much weaker than the unassembled or partially assembled dystrophin fragments. Varying the ratio of the three plasmids improved the relative abundance of the FL versus the unassembled or partially assembled dystrophin signals, with the 4:2:1 ratio of N1:M1:C1 plasmids yielding the highest level of FL-dystrophin. The C-terminal fragment band was much more intense than either the FL or MC bands, indicating that the assembly between the M and C fragments mediated by Gp41-1 intein needs to be improved.

### Optimization of split intein constructs to improve FL-dystrophin assembly

Encouraged by this initial observation, we attempted to optimize the assembly between the M and C fragments. First, we tested a different split-site (IGA-SPT) within the H3 domain and mutated the −1 position from alanine to tyrosine to favor the Gp41-1-mediated PTS (Fig. 2a). These changes (Dys-M2 and Dys-C2) led to a substantial improvement in the FL-dystrophin assembly (Fig. 2b, c) with reduced unassembled (Fig. 2d–f) and partially assembled fragments (Supplementary Fig. 1a–c). In addition, we removed a small intron within the C fragment construct (Dys-C3), which significantly increased the FL-dystrophin band intensity (Fig. 2b, c). Next, we reasoned that the +1 to +3 position on the extein (e.g., Dys-C3) may also affect the PTS efficiency. We thus mutated the SPT sequence to SSS at the junction site on the C fragment construct (Fig. 2g). In addition, we changed the strong Kozak sequence to a weak one considering the abundance of the C fragment. However,

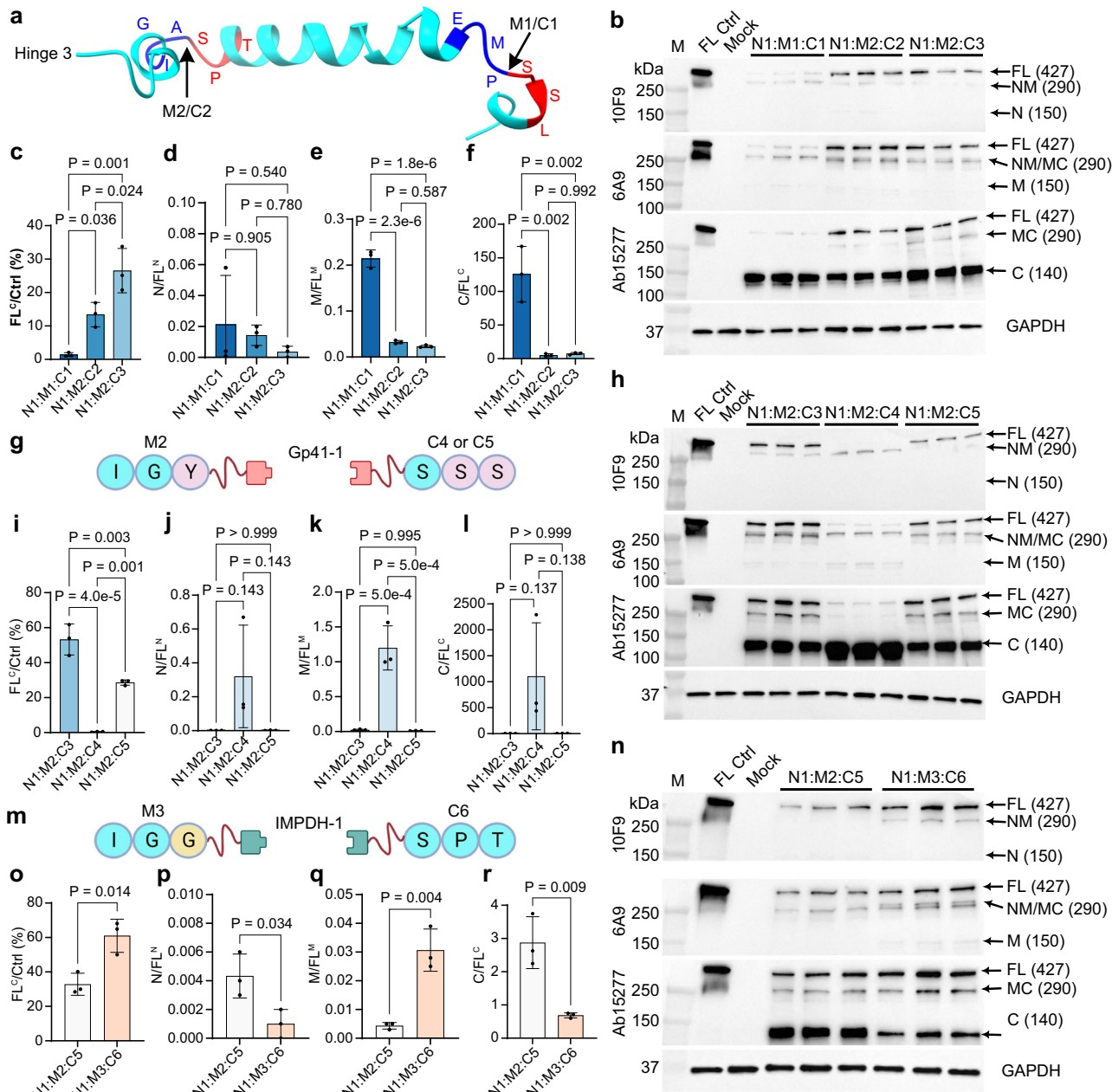

**Fig. 2 | Optimization of split intein constructs to assemble FL-dystrophin. a** The predicted structure of the Hinge 3 domain with the two split sites marked by the arrows. The three residues at each side of the split sites were labeled in blue and red. **b, h, n** Western blotting analysis of dystrophin expression in HEK293 cells transfected with or without different versions of Dys-N, M, and C constructs at a molar ratio of 4:2:1. HEK293 cell lysate transfected with a FL-dystrophin construct was used as a positive control (FL Ctrl) and the GAPDH was used as a loading control. **c–f, i–l, o–r** Densitometry quantification of the FL-dystrophin band intensity (**c, i, o**), the ratio of unassembled N versus FL (**d, j, p**), the ratio of unassembled M versus FL (**e, k, q**) and the ratio of unassembled C versus FL (**f, l, r**) from three biological repeats per condition. One-way ANOVA with Tukey's multiple comparisons test for three groups and two-tailed unpaired Student's *t*-test for two groups. **g, m** Diagrams showing the M and C constructs at the split sites with the −1 to −3 and +1 to +3 residues in the dystrophin exteins labeled. The mutated amino acids are labeled in purple (**g**) and yellow (**m**). Data were mean ± SEM. Source data are provided as a Source Data file.

these two changes (Dys-C4) led to a substantial reduction of the FL-dystrophin band with an increased accumulation of the NM band and C band (Fig. 2h–l and Supplementary Fig. 1d–f). We reasoned that this might be caused by the use of the weak Kozak sequence, which potentially leads to the translation initiation from a downstream in-frame start codon so that the C fragment was expressed without the intein fusion. Indeed, after we switched the weak Kozac sequence back to a strong one in this version of the C fragment construct (Dys-C5), FL-dystrophin expression was restored (Fig. 2h–l and Supplementary Fig. 1d–f). Of note, we also added a synthetic signal for ubiquitin-

dependent proteolysis (PB29)[34] in the Dys-C5 construct to lower the expression level of the C fragment. Although the FL-dystrophin band signal was reduced when compared to Dys-C3, we observed that the NM fragment was almost undetectable, suggesting that these changes improved NM and C assembly. We further tested a different intein (IMPDH−1), which has a fast PTS rate comparable to Gp41-1 with a native junction sequence of GGG-SIC[33], similar to the split site of dystrophin (IGA-SPT) (Fig. 2m). We also mutated alanine at the -1 position of Dys-M2 extein to glycine to further mimic the native junction sequence of IMPDH-1. These changes (Dys-M3 and Dys-C6)

significantly improved the FL-dystrophin signal by ~86% (Fig. 2n, o). Moreover, the unassembled C fragment band was dramatically reduced by ~76% as compared to Dys-N1/M2/C5 (Fig. 2n, r), with marginal effects on the unassembled N and M fragments (Fig. 2n, p, q) and partially assembled fragments (Supplementary Fig. 1g–i). Finally, we inserted two different poly-adenylation signal sequences[35] at the upstream of the start codon in order to further lower the expression level of the C fragment (Dys-C7 and Dys-C8) (Supplementary Fig. 2a). These changes significantly reduced the C-fragment band intensity while still maintaining a high level of FL-dystrophin expression (Supplementary Fig. 2b, c, f) and similar N/FL (Supplementary Fig. 2d) or MC/FL ratio (Supplementary Fig. 2i). However, we found that lowering C-fragment expression further caused a concomitant accumulation of the M (Supplementary Fig. 2e) and NM (Supplementary Fig. 2g, h) fragments. These data suggest that some excess of the C-fragment favors the assembly of FL-dystrophin.

In contrast to the canonical inteins, atypical inteins with small intein[N] and large intein[C] have also been discovered[36,37]. To test whether the atypical inteins could also enable the assembly of FL-dystrophin, we modified the N, M, and C constructs using the atypical inteins Cat[38] and VidaL[39] to mediate the N-M and M-C splicing, respectively (Supplementary Fig. 3a). Western blotting analysis showed that these atypical inteins can also confer the assembly of FL-dystrophin, but the efficiency was much lower than Dys-N1/M3/C6 (Supplementary Fig. 3b). We chose the best-optimized version Dys-N1/M3/C6 for further in vivo studies.

### Restoration of FL-dystrophin expression in $mdx^{4cv}$ mice following systemic MyoAAV delivery of Dys-N1/M3/C6

We first replaced the generic promoter meCMV with a synthetic muscle-specific promoter Spc5-12[40] (for the N and M fragment construct) or Spc2-26[40] (for the C fragment construct) for AAV packaging. We chose a recently engineered myotropic AAV capsid, MyoAAV4A, for packaging because of its superior muscle and heart transduction in mice and monkeys following systemic delivery[31]. A total dose of 2E + 14 vg/kg AAV vectors consisting of N1, M3, and C6 at a 2:1:1 molar ratio was delivered into a cohort of $mdx^{4cv}$ mice ($N = 13$) at the age of 3–4 weeks via retro-orbital injection (Fig. 3a).

Immunofluorescence staining was performed using the aforementioned anti-dystrophin antibodies (Fig. 3b). Dystrophin expression could be detected at the sarcolemma of wild-type (WT) *gastrocnemius* (GA) muscle with all these three antibodies, while GA muscle from $mdx^{4cv}$ mice was negative for any of these antibody stains (Fig. 3b). AAV administration restored dystrophin expression that can be detected by all these three antibodies (Fig. 3b). Importantly, dystrophin signals were correctly localized at the sarcolemma without noticeable accumulation in the cytoplasm. On average, dystrophin was detected in 83.4 ± 2.7% muscle fibers (Fig. 3c). Dystrophin expression was also robustly rescued in the cardiac muscles of $mdx^{4cv}$ mice with 78.3 ± 2.7% cardiomyocytes being dystrophin positive following AAV administration (Supplementary Fig. 4a, c). However, dystrophin+ fibers in diaphragm muscles (8.6 ± 2.6%) were much lower than those in the GA muscles (Supplementary Fig. 4b, d).

Western blotting was also performed to substantiate these observations. Again, FL-dystrophin was readily detectable using the three different N-, M-, or C-recognizing antibodies in GA muscles from $mdx^{4cv}$ mice treated with AAV-N1/M3/C6 (Fig. 3d). Interestingly, both the N- and M-recognizing antibodies detected mostly the FL-dystrophin with weak partially assembled dystrophin fragments, while the unassembled N- or M-fragment was hardly discernable (Fig. 3d). The C-recognizing antibody detected FL-dystrophin and unassembled C-fragment with roughly equal intensities, while the partially assembled MC fragment was almost undetectable (Fig. 3d). FL-dystrophin expression was also readily detectable in the heart muscles of AAV-treated $mdx^{4cv}$ mice (Supplementary Fig. 4e), but the diaphragm muscle showed a much weaker expression of FL-dystrophin and the C fragment following AAV treatment (Supplementary Fig. 4f), likely reflecting weak activity of the Spc2-26 promoter and/or MyoAAV4A capsid in diaphragm muscle.

The loss of dystrophin in dystrophic muscle severely affects the integrity of the entire DGC[5–7]. To test if AAV-N1/M3/C6 treatment restores the other components of the DGC in $mdx^{4cv}$ muscles, we performed immunofluorescence staining with the antibodies against various components of the DGC such as α-sarcoglycan (α-SG), β-SG, α-dystroglycan (α-DG), β-DG, neuronal nitric oxide synthase (nNOS), and α-dystrobrevin (α-DB). As shown in Fig. 3e, the DGC components, including α-SG, β-SG, α-DG, β-DG, nNOS, and α-DB, were all severely reduced at the sarcolemma of GA muscle fibers from $mdx^{4cv}$ mice but were substantially restored by AAV-N1/M3/C6 treatment.

### Functional and histopathological improvement in $mdx^{4cv}$ mice following systemic MyoAAV4A delivery of Dys-N1/M3/C6

Increased muscle injury and reduced muscle force production are the pathological hallmarks of DMD. To examine if the AAV-N1/M3/C6 treatment improves the muscle pathologies in $mdx^{4cv}$ mice, we first measured the serum creatine kinase (CK) levels at 5 weeks following AAV administration. As compared to WT mice, $mdx^{4cv}$ animals showed a dramatic elevation in serum CK (WT: 220.6 ± 109.1, $n = 14$ vs $mdx^{4cv}$: 3946.0 ± 341.1, $n = 12$; $p < 0.0001$, Fig. 4a), which was significantly reduced in AAV-treated group (1060.0 ± 229.4, $n = 13$; $p < 0.0001$), suggesting that FL-dystrophin expression reduces muscle injury in dystrophic mice. To test if AAV-N1/M3/C6 treatment improves muscle function, we measured the muscle contractility using an in vivo muscle test system[28,41,42]. The maximum plantarflexion tetanic torque was measured during supramaximal electric stimulation of the tibial nerve at 150 Hz. As previously shown, $mdx^{4cv}$ mice produced greatly reduced torque as compared to WT controls (WT: 544.7 ± 9.8, $n = 8$ vs $mdx^{4cv}$: 298.2 ± 10.6, $n = 10$; $p < 0.0001$, Fig. 4b). Systemic delivery of AAV-N1/M3/C6 significantly increased the tetanic torque in $mdx^{4cv}$ mice by ~51.7% (452.5 ± 12.0, $n = 11$; $p < 0.0001$, Fig. 4b).

To examine if AAV-N1/M3/C6 treatment improves the histopathology of $mdx^{4cv}$ mice, we performed Hematoxylin and Eosin (H&E) staining of skeletal muscle sections from the animals. While WT GA muscle sections showed normal musculature, $mdx^{4cv}$ mice displayed a typical muscular dystrophy phenotype as evidenced by the presence of centrally nucleated muscle fibers (CNFs), muscle necrosis, and regeneration. These pathologies were substantially ameliorated by AAV-N1/M3/C6 administration (Fig. 4c). To further quantify the percentages of CNFs, we performed immunofluorescence staining of the muscle sections with anti-laminin α2 and 4′,6-diamidino-2-phenylindole (DAPI) (Fig. 4c). The CNFs in the GA muscles of $mdx^{4cv}$ mice were reduced from 58.2 ± 1.7% to 25.1 ± 1.5% by AAV-N1/M3/C6 treatment (Fig. 4d). Owing to the repeated cycles of degeneration and regeneration, the distribution of muscle fiber size in $mdx^{4cv}$ GA shifted to lower sizes as compared to WT (Fig. 4e), whereas AAV-N1/M3/C6 treatment shifted the fiber size distribution towards those of the WT muscles (Fig. 4e). It appears that AAV-N1/M3/C6 treatment also improved the histopathology of $mdx^{4cv}$ diaphragm muscles (Supplementary Fig. 5a), but we did not observe significant changes in CNFs in the diaphragm muscles (Supplementary Fig. 5b), consistent with the low dystrophin restoration in AAV-treated $mdx^{4cv}$ diaphragm muscles. To examine the impact of AAV-N1/M3/C6 treatment on fibrosis, we performed Masson's Trichrome staining on muscle sections, which showed that the fibrosis in both GA (Fig. 4c, f) and diaphragm (Supplementary Fig. 5c, d) muscles of $mdx^{4cv}$ mice was greatly attenuated by AAV-N1/M3/C6 treatment.

### Effects of promoter and dose on FL-dystrophin expression in skeletal and heart muscles of $mdx^{4cv}$ mice

We reasoned that the Spc2-26 promoter may not work efficiently in the diaphragm muscle, thus yielding low restoration of FL-dystrophin in

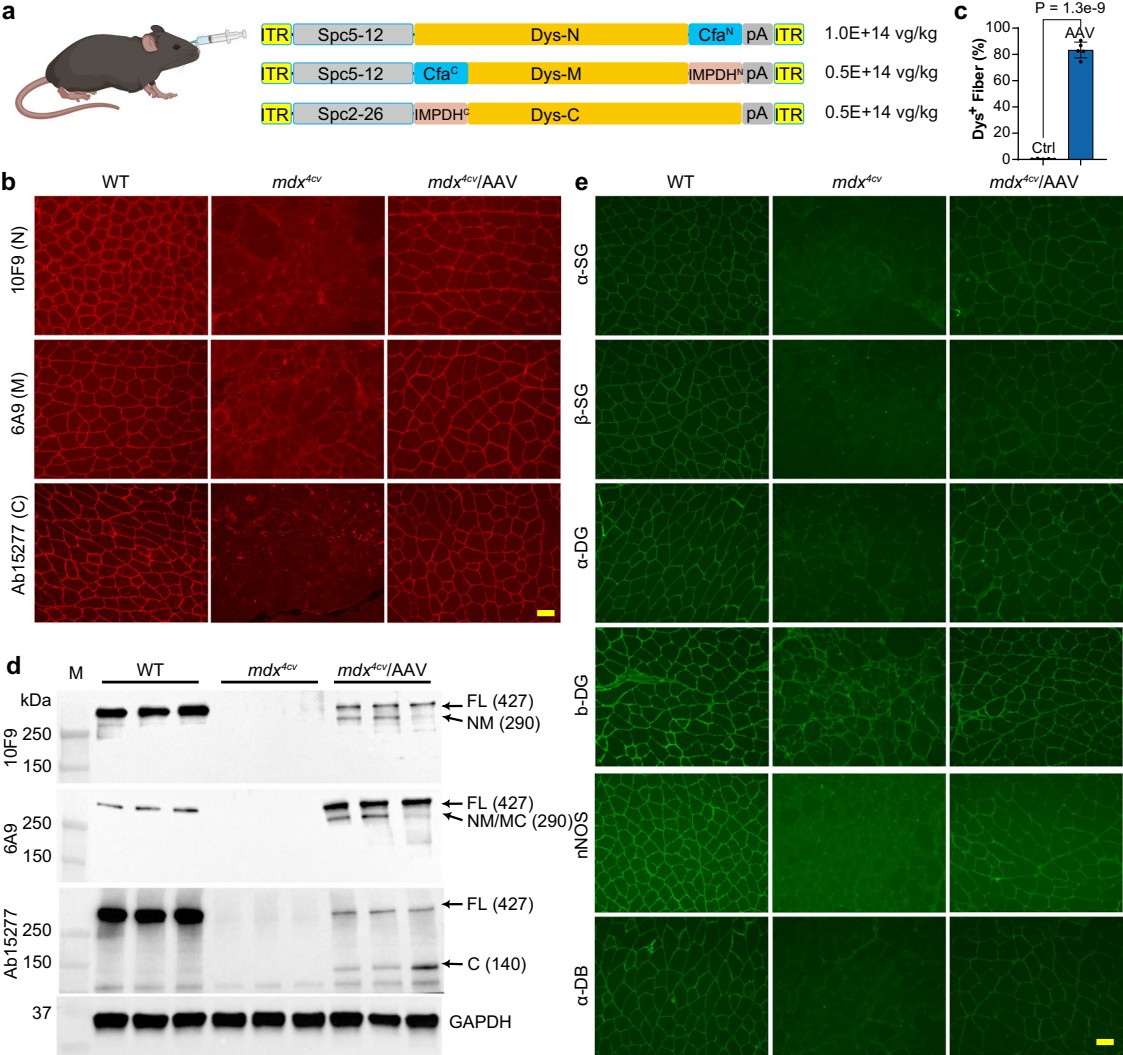

**Fig. 3 | Restoration of FL-dystrophin expression in *mdx^{4cv}* mice following systemic MyoAAV4A delivery of Dys-N1/M3/C6.** All animal experiments were performed in WT and *mdx^{4cv}* male mice with the C57BL/6J genetic background. **a** The diagram depicting retro-orbital injection of MyoAAV4A carrying Dys-N1/M3/C6, the three MyoAAV4A constructs, and their respective doses administered. **b** Immunofluorescence images of GA muscle sections from WT and *mdx^{4cv}* mice treated with or without AAV (10 weeks of age, *n* = 5 per group), stained with three different anti-dystrophin antibodies recognizing N, M, and C fragments, respectively. Scale bar: 50 μm. **c** Quantification of dystrophin-positive muscle fibers in GA muscles of 10-week-old *mdx^{4cv}* mice treated with or without AAV (*n* = 5 per group, two-tailed unpaired Student's *t*-test). **d** Western blotting analysis of dystrophin expression in GA muscles from WT and *mdx^{4cv}* mice treated with or without AAV (10 weeks of age, *n* = 3 per group). **e** Immunofluorescence staining of GA muscle sections from WT and *mdx^{4cv}* mice treated with or without AAV (10 weeks of age, *n* = 5 per group) using the antibodies against the DGC components including α-SG, β-SG, α-DG, β-DG, nNOS, and α-DB. Scale bar: 50 μm. Data were mean ± SEM. Source data are provided as a Source Data file. **a** created with BioRender.com released under a Creative Commons Attribution-NonCommercial-NoDerivs 4.0 International license (https://creativecommons.org/licenses/by-nc-nd/4.0/deed.en).

this tissue. To test this, we changed the promoter in the C construct to Spc5-12, and designated AAV-N1/M3/C6-Spc5-12 as AAV-FL-v2 and the original AAV-N1/M3/C6 as AAV-FL-v1. Both AAV mixtures (a total dose of 2E + 14 vg/kg at 2:1:1 ratio for N, M, and C vectors) were systemically administered into a cohort of *mdx^{4cv}* mice via retro-orbital injection. At 6 weeks post AAV injection, the animals were sacrificed for examination.

Western blotting analysis showed that the expression of FL-dystrophin was comparable between AAV-FL-v1 and AAV-FL-v2 in GA muscles (Fig. 5a–d). To estimate the relative amount of rescued FL-dystrophin compared to endogenous dystrophin in healthy skeletal muscle, control human skeletal muscle lysate was loaded at 50, 25, and 10% on the gel. Three different antibodies recognizing the N, M, or C

fragment reported 32.6–49.5% and 36.9–43.4% of FL-dystrophin restoration following AAV-FL-v1 and AAV-FL-v2 treatment, respectively (Fig. 5b–d). In diaphragm muscles, AAV-FL-v2 treatment significantly increased FL-dystrophin expression when compared to AAV-FL-v1 as detected by anti-M and C antibodies (Fig. 5e–h). Similar improvement of FL-dystrophin rescue was observed in heart muscles (Fig. 5i–l), indicating that the Spc5-12 promoter works more efficiently in the diaphragm and heart muscles than Spc2-26. It is of note that we used human skeletal muscle lysate to estimate the relative amount of FL-dystrophin rescue for both diaphragm and heart muscles, as we do not have access to human diaphragm and heart muscle samples.

As the total dosage of MyoAAV that we used was relatively high (2E + 14 vg/kg), we wondered whether a lower total dosage

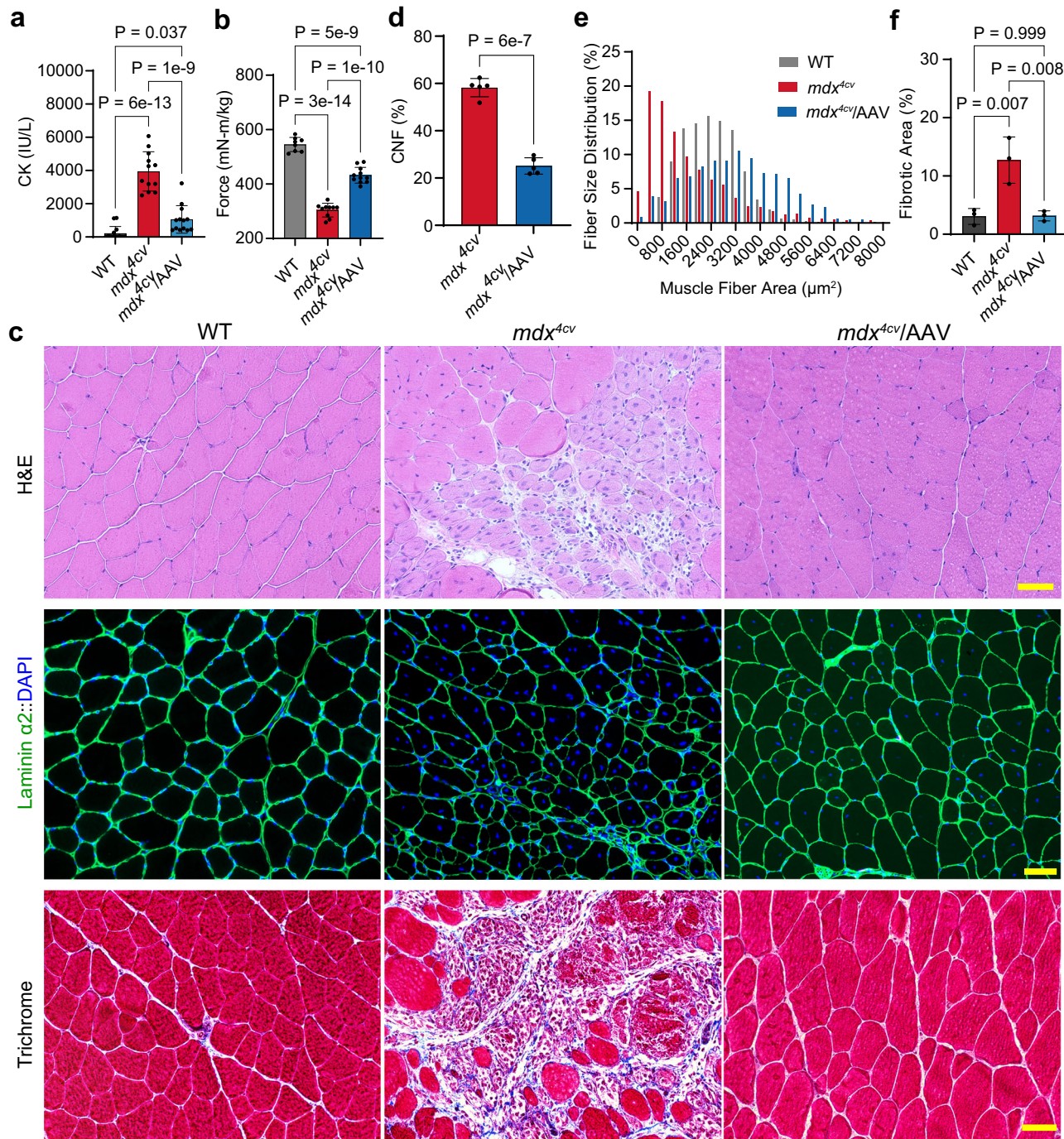

**Fig. 4 | Functional and histopathological improvement in _mdx^4cv_ mice following systemic MyoAAV4A delivery of Dys-N1/M3/C6.** All animal experiments were performed in WT and _mdx^4cv_ male mice with the C57BL/6J genetic background. **a** Measurement of serum CK in 8-week-old mice (_n_ = 14 WT, 12 _mdx^4cv_ and 13 AAV-treated _mdx^4cv_ mice). **b** Tetanic torque measurements of the posterior compartment muscles of the mice (_n_ = 8 WT, 10 _mdx^4cv_ and 11 AAV-treated _mdx^4cv_; 8–12 weeks of age). **c** H&E staining, immunofluorescence staining with anti-laminin α2/DAPI and Masson's Trichrome staining images of GA muscle sections (_n_ = 5 per group; 10 weeks of age). Scale bar: 50 μm. **d** Measurement of CNF in the GA muscles of 10-week-old _mdx^4cv_ mice with or without AAV treatment (_n_ = 5 each, two-tailed unpaired _t_-test). **e** Muscle fiber size distribution in GA muscles of the mice at 10 weeks of age (_n_ = 5 WT, 5 _mdx^4cv_, and 6 AAV-treated _mdx^4cv_). **f** Quantitative analysis of fibrotic area in the GA muscle sections of the mice at 10 weeks of age (_n_ = 3 per group). Data were mean ± SEM. Statistical analyses were performed by one-way ANOVA with Tukey's multiple comparisons test except otherwise specified. Source data are provided as a Source Data file.

(8E + 13 vg/kg at 2:1:1 of N, M, and C) could yield efficient FL-dystrophin restoration in _mdx^4cv_ mice. As shown in Fig. 5a–d, FL-dystrophin was readily detectable in GA muscles treated with the lower dose, albeit at a lower level as compared to the high dose group. The differences between the low and high dose groups were less evident in the diaphragm muscle (Fig. 5e–h) and heart (Fig. 5i–l).

## Comparison of MyoAAV-delivered FL- and micro-dystrophin in _mdx^4cv_ mice

Finally, to benchmark against micro-dystrophin gene therapy, we performed a comparative study for MyoAAV-delivered FL- and micro-dystrophin gene delivery in _mdx^4cv_ mice. Two different micro-dystrophin constructs were tested with one like the Pfizer version but containing the fusion of spectrin repeat (SR) 2 and 22 (designated

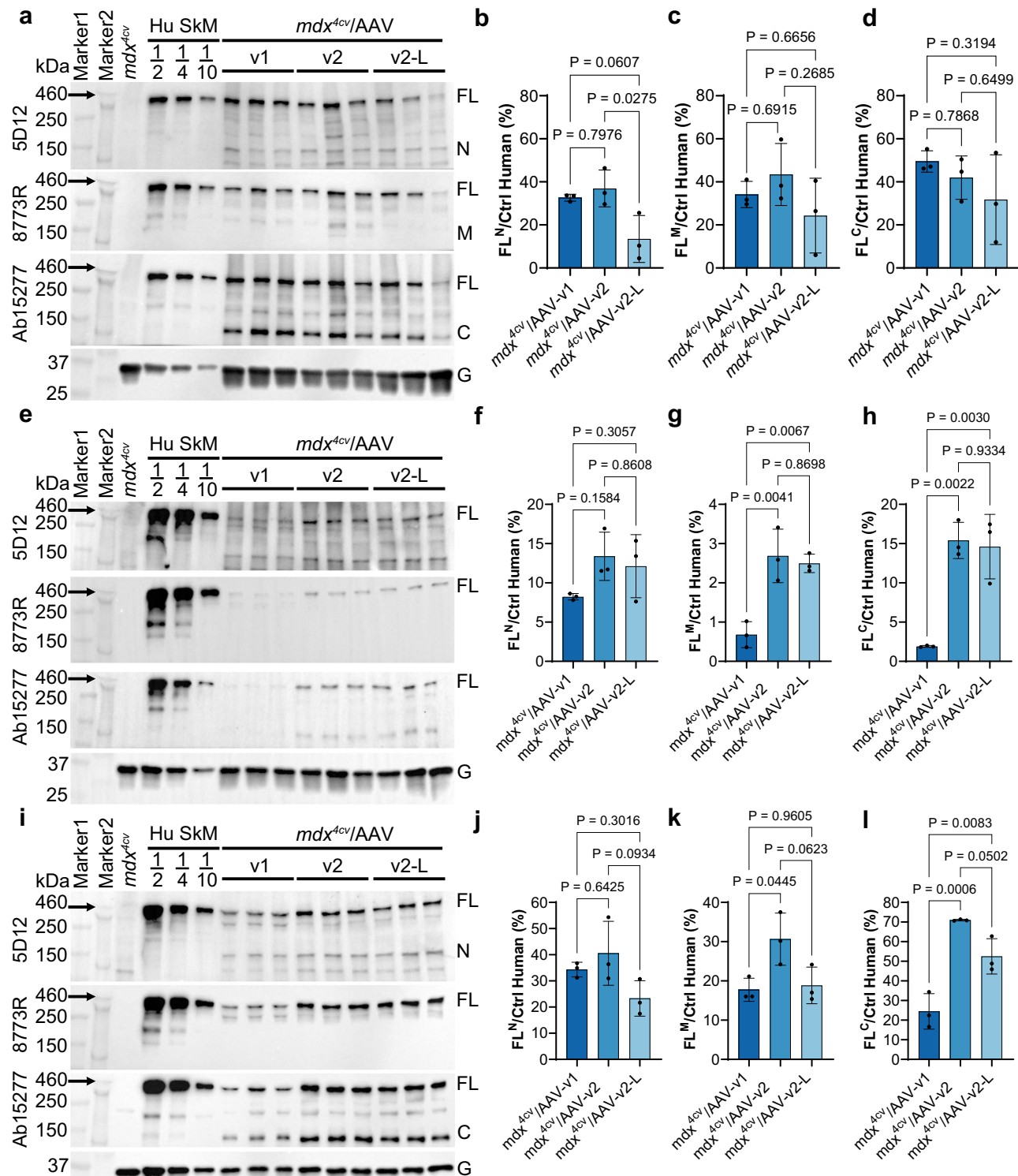

**Fig. 5 | Effects of promoter and dose on FL-dystrophin expression in skeletal and heart muscles of *mdx^4cv* mice.** All animal experiments were performed in WT and *mdx^4cv* male mice with the C57BL/6J genetic background. **a–l** Western blotting (**a**, **e**, **i**) and quantification (**b–d**, **f–h**, **j–l**) of dystrophin expression in GA (**a–d**), diaphragm (**e–h**), and heart (**i–l**) muscles from WT and *mdx^4cv* mice treated with or without AAV-FL-v1 or AAV-FL-v2 at 2E + 14 vg/kg (v1, v2) or 8E + 13 vg/kg (AAV-FL-v2-

L) (10 weeks of age, *n* = 3 per group). Control human skeletal muscle lysate was loaded at 50% (1/2), 25% (1/4), or 10% (1/10). FL^N, FL^M, and FL^C denoted the FL-dystrophin band quantification using blots detected by different antibodies against N, M, and C fragments, respectively. Data were mean ± SEM. Statistical analyses were performed by one-way ANOVA with Tukey's multiple comparisons test except otherwise specified. Source data are provided as a Source Data file.

as µ-v1), and the other originally developed in Duan's laboratory (designated as µ-v2) (Fig. 6a). All the dystrophin constructs were under the control of the Spc5-12 promoter. AAV-µ-v1 and µ-v2 were delivered at 8E + 13 vg/kg, while the total dose of AAV-FL-v2 were 2E + 14 or

8E + 13 vg/kg (an effective dose of 5E + 13 or 2E + 13 vg/kg determined by the lowest dose among the three fragments). Consistent with the data shown in Fig. 5, FL-dystrophin was restored to 46.3% of normal human skeletal muscle level in GA muscles after AAV-FL-v2 delivery at

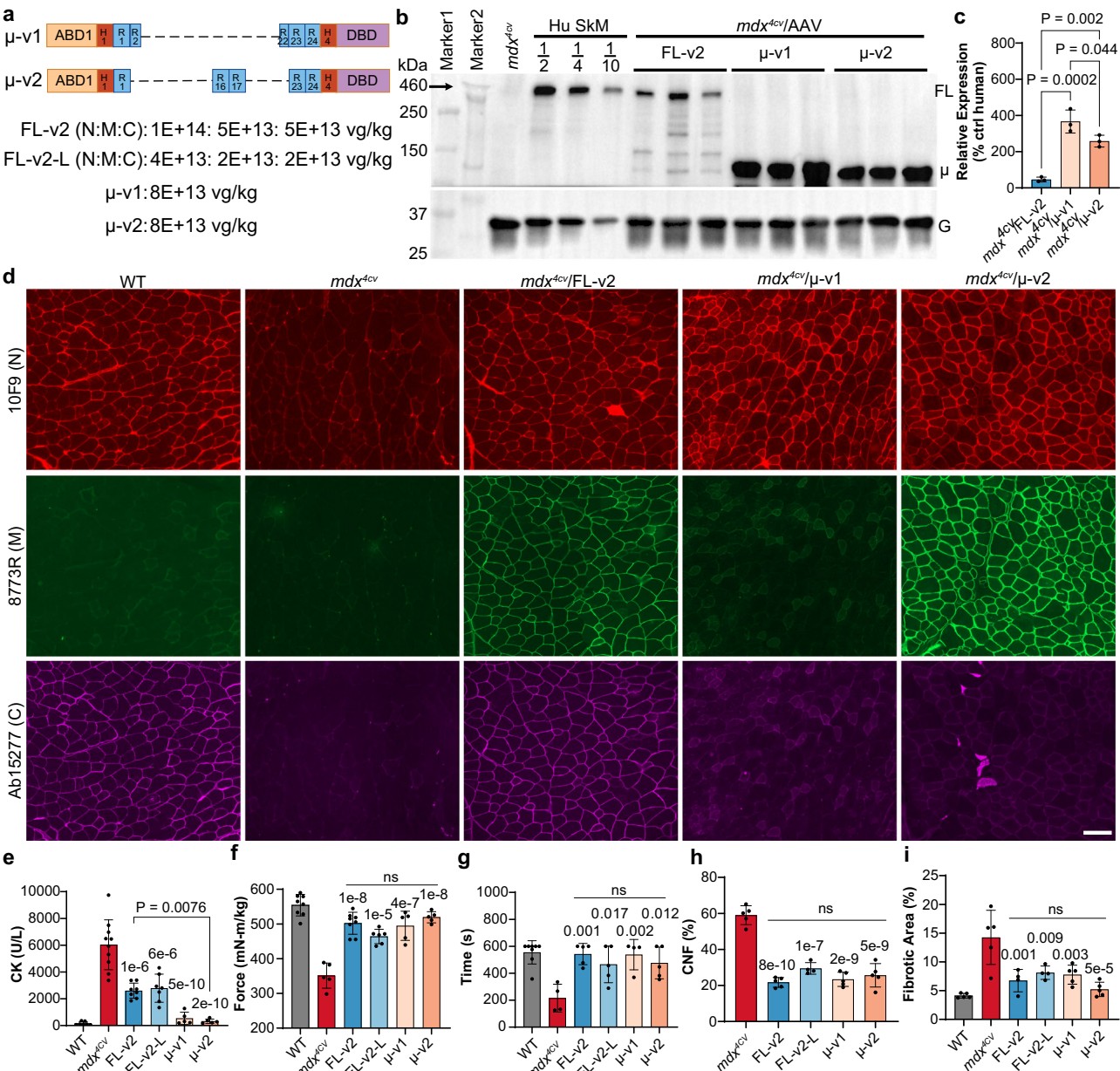

**Fig. 6 | Comparison of MyoAAV-delivered FL- and micro-dystrophin in *mdx^{4cv}* mice.** All animal experiments were performed in WT and *mdx^{4cv}* male mice with the C57BL/6J genetic background. **a** Diagram illustrating the micro-dystrophin constructs and their doses used in this study. **b**, **c** Western blotting and quantification of dystrophin expression in GA muscles of *mdx^{4cv}* mice treated with or without AAV-FL-v2, AAV-µ-v1, or AAV-µ-v2 ($n = 3$ mice per group). Control human skeletal muscle lysate was loaded at 50% (1/2), 25% (1/4), or 10% (1/10). Only the N-recognizing antibody (5D12) was tested as the micro-dystrophins do not have the immunogen for C or both M and C. **d** Co-immunofluorescence staining with anti-dystrophin antibodies (N: 5D12 + M: 8773 R; N: 5D12 + C: 15277) in consecutive sections of GA muscles ($n = 3$ mice per group). Scale bar: 100 µm. Note: the rabbit polyclonal antibody 8773R recognizing SR16/17 works for human, but not mouse, dystrophin. **e** Measurement of serum CK in 8-week-old mice ($n = 7$ WT, 10 *mdx^{4cv}*, 8

AAV-FL-v2, 7 AAV-FL-v2-L, 5 AAV-µ-v1, and 5 AAV-µ-v2 treated *mdx^{4cv}*). **f** Tetanic torque measurements of the posterior compartment muscles of the mice ($n = 8$ WT, 5 *mdx^{4cv}*, 8 AAV-FL-v2, 6 AAV-FL-v2-L, 5 AAV-µ-v1, and 5 AAV-µ-v2 treated *mdx^{4cv}*; 9 weeks of age). **g** Measurement of the hanging time before the mice fell from the wire mesh ($n = 8$ WT, 4 *mdx^{4cv}*, 5 AAV-FL-v2, 5 AAV-FL-v2-L, 4 AAV-µ-v1, and 5 AAV-µ-v2 treated *mdx^{4cv}*; 9–10 weeks of age). **h** Measurement of CNF in the GA muscles of 10-week-old *mdx^{4cv}* mice with or without AAV treatment ($n = 5$ *mdx^{4cv}*, 5 FL-v2, 4 FL-v2-L, 5 µ-v1, and 5 µ-v2 treated *mdx^{4cv}*). **i** Quantitative analysis of fibrotic area in GA muscle sections of the mice at 10 weeks of age ($n = 5$ WT, 5 *mdx^{4cv}*, 5 AAV-FL-v2, 4 AAV-FL-v2-L, 5 AAV-µ-v1, and 5 AAV-µ-v2 treated *mdx^{4cv}*). Data were mean ± SEM. Statistical analyses were performed by one-way ANOVA with Tukey's multiple comparisons test except otherwise specified. Source data are provided as a Source Data file.

the high dose group (Fig. 6b, c), however, the micro-dystrophin proteins were overexpressed at 3.7 and 2.6 folds following AAV-µ-v1 and µ-v2 delivery, respectively (Fig. 6b, c). Similarly, AAV-FL-v2 delivery led to the restoration of FL-dystrophin to 10.0% in the diaphragm and 69.3% in the heart using human skeletal muscle as a reference, while micro-dystrophin was overexpressed at 2.6–7.8 folds in diaphragm and heart following AAV-µ-v1 and µ-v2 delivery (Supplementary Fig. 6a–d).

To examine the expression of FL-dystrophin and micro-dystrophin proteins in muscle fibers, we performed co-immunofluorescence staining with anti-N (10F9, which recognizes the hinge1 region) and anti-M (8773R, which recognizes the SR16/17 of human dystrophin only) on one section, and co-immunofluorescence staining with anti-N and anti-C (Ab15277, which recognizes the C-terminus) on a consecutive section of GA muscles. As shown in Fig. 6d,

WT skeletal muscle was positive for both N and C antibodies but not for M antibody as expected, and $mdx^{4cv}$ skeletal muscle showed only background signal for either of these antibodies. AAV-μ-v1 yielded positive staining for only N antibody, while AAV-μ-v2 led to positive staining for both N and M antibodies. The $mdx^{4cv}$ mice treated with AAV-FL-v2 were positive for all three antibodies, and the signals were correctly localized at the sarcolemma. Interestingly, visual examination of the fluorescence images could rarely find any muscle fibers positively for only one or two antibodies (e.g., if a muscle fiber was positive for one antibody, it was also positive for the other two antibodies) in $mdx^{4cv}$ mice treated with AAV-FL-v2. To further illustrate this high degree of correlation among the three antibody signals, we performed line profile analyses on these images for AAV-FL-v2 treated mice. As shown in Supplementary Fig. 7, the N and M immunofluorescence signals showed almost identical patterns at three arbitrarily selected lines. Similarly, the N and C immunofluorescence signals also showed highly overlapping peaks along the lines. These results suggest that muscle fibers expressing only the unassembled or partially assembled dystrophin fragments are very rare, if any.

The serum CK levels were significantly decreased in all AAV-treated groups, with the μ-v1 and μ-v2 groups approaching the WT levels (Fig. 6e). Muscle contractility was significantly increased in all treatment groups compared to control $mdx^{4cv}$ mice, and there was no significant difference among the treatment groups (Fig. 6f). We further performed a wire hanging test to evaluate the overall muscle strength in these mice. The latency to when the animal falls was recorded and compared for the animals in each group. On average, WT mice stayed on the wire mesh for over $555.1 \pm 30.4$ s ($n = 8$), while $mdx^{4cv}$ mice held only for $215.4 \pm 51.6$ s ($n = 4$; $p = 0.0002$, Fig. 6g). Remarkably, all AAV treatments completely normalized the hanging time on the wire mesh (AAV-FL-v2: $542.1 \pm 35.2$ s, $n = 5$; AAV-FL-v2-L: $466.0 \pm 60.8$ s, $n = 5$; AAV-μ-v1: $538.0 \pm 56.2$ s, $n = 4$; AAV-μ-v2: $476.6 \pm 52.6$ s, $n = 5$; Fig. 6g).

Histological examination showed that the muscular dystrophy features were substantially ameliorated in the GA muscles from all AAV treatment groups (Supplementary Fig. 8a). The percentage of CNF was decreased from $59.0 \pm 2.3$ in $mdx^{4cv}$ to $21.7 \pm 1.2$, $29.5 \pm 1.6$, $23.4 \pm 1.8$, and $25.7 \pm 2.9$ in the AAV-FL-v2, AAV-FL-v2-L, AAV-μ-v1, and AAV-μ-v2 treatment groups, respectively (Fig. 6h). Fibrotic area was also significantly decreased in AAV treatment groups with no difference observed among them (Fig. 6i). Similar histopathological improvement was observed in diaphragm (Supplementary Fig. 8b). Histological examination of heart sections showed no significant differences among all groups (Supplementary Fig. 8c).

Although the histopathological and functional assays showed similar improvement for AAV-FL-v2 and micro-dystrophins, previous studies showed that micro-dystrophins failed to correct cavin-associated ERK signaling defects in dystrophic mouse hearts due to the lack of dystrophin's C-terminal domain[43]. In agreement with that, our data showed that the membrane localization of cavin-4 is disrupted in cardiomyocytes of $mdx^{4cv}$ mice and AAV-delivered micro-dystrophins did not correct such mis-localization of cavin-4 (Fig. 7a). However, AAV-delivered FL-dystrophin greatly restored the membrane localization of cavin-4 in $mdx^{4cv}$ cardiomyocytes (Fig. 7a). In response to cardiac stress/damage, membrane-associated cavin-4 recruits the signaling molecule ERK to caveolae to activate key cardio-protective responses. Western blot analysis showed that ERK phosphorylation was inhibited in $mdx^{4cv}$ mouse heart, which was not affected by micro-dystrophin gene delivery, but was significantly improved by FL-dystrophin gene delivery (Fig. 7b–d). Taken together, these data suggest that FL-dystrophin gene therapy is superior to micro-dystrophin gene therapy.

## Discussion

The small packaging capacity of AAV vectors limits gene replacement therapy for DMD and many other diseases. Until now, delivering full-length dystrophin to bodywide muscles has been unsuccessful. Here we harnessed the split intein-mediated PTS to develop a triple AAV system for efficient assembly and in vivo delivery of FL-dystrophin. Using the engineered myotropic AAV capsid, we achieved systemic FL-dystrophin rescue in both skeletal muscle and heart, which led to significant functional, histopathological, and biochemical signaling improvement in $mdx^{4cv}$ mice.

Using two orthogonal split inteins (Cfa and Gp41-1 or IMPDH-1) with a very fast rate of PTS, we demonstrate the feasibility of generating FL-dystrophin protein in vitro and in vivo. After a series of optimizations for the construct design (e.g. the choice of split inteins, the split sites, the addition of a degradation signal, mutation of the junctional amino acids, etc.), we gradually improved the assembly efficiency. The best version (N1/M3/C6) can achieve ~60% of FL-dystrophin protein expression (relative to a plasmid carrying the entire dystrophin cDNA) with low levels of unassembled or partially assembled products in vitro. Earlier studies with $Ssp$ DnaB split intein to assemble a 6.3-kb Becker-form dystrophin[44] and factor VIII[45] were inefficient. These results showed that the selection of split inteins and split sites on dystrophin (and thus the junctional amino acids) are particularly important for efficient FL-dystrophin assembly. Future efforts could systemically screen different orthogonal pairs of split inteins to further increase the assembly efficiency, expression, and purity of fully assembled FL-dystrophin protein. In addition, our optimized triple vector system can be easily adapted to deliver large midi-dystrophin proteins using dual vectors.

The split site for N and M constructs is in proximity to the natural initiation site of the Dp260 isoform, whereas the split site for M and C constructs overlaps with the start site of the Dp140 isoform. The selection of these split sites may help minimize the potential negative impacts of unassembled C and partially assembled MC fragments. A previous study showed that the Dp260 (MC fragment) restored a stable association between costameric actin and the sarcolemma, assembled the DGC, and significantly slowed the progression of muscular dystrophy in $mdx$ mice[46]. Metzger and colleagues showed that the NtermDys fragment (corresponding to our NM fragment) did not compete with dystrophin and had no pathological effect, but the 2A protease-cleaved CtermDys (corresponding to our unassembled C fragment) was sufficient to cause dystrophic cardiomyopathy in transgenic mice when overexpressed[47]. However, in their CtermDys transgenic mice, the CtermDys was overexpressed by more than 10 folds relevant to the FL-dystrophin (estimated from Fig. 1b in ref. 47), while in our AAV-delivered mice, the unassembled C fragment was about equal to the FL-dystrophin product (Fig. 5i). Thus, it is very unlikely that the unassembled C fragment would cause cardiomyopathy in our triple vector delivery. In agreement with this, we did not observe increased cardiac fibrosis in $mdx^{4cv}$ mice following the triple vector delivery (Supplementary Fig. 8c). Nevertheless, it is important to carefully investigate the long-term impact of these unassembled and partially assembled products in the future.

We optimized the molar ratios for each vector carrying the N, M, and C fragments by semi-quantitative assessment of each fragment and the resulting assembly products by Western blotting initially in vitro. The N and M fragments appeared to be less stable than the C fragment. We observed that the expression level of the N fragment fused with $Cfa^N$ was evidently lower than that fused with the smaller $Cat^N$, indicating that the larger $Cfa^N$ fused at the C-terminus of the N construct contributed to the instability of the N fragment fusion protein[48]. Although our efforts in this work were primarily centered on optimizing the inteins and split sites as well as tuning down the C fragment expression, future studies can be targeted to improve the stability of the N and M fragments, which may improve the expression level of FL-dystrophin. A few strategies can be explored. First, codon optimization could potentially help to boost the N and M transgene expression. Second, adding an intein "cage" at the end may help to

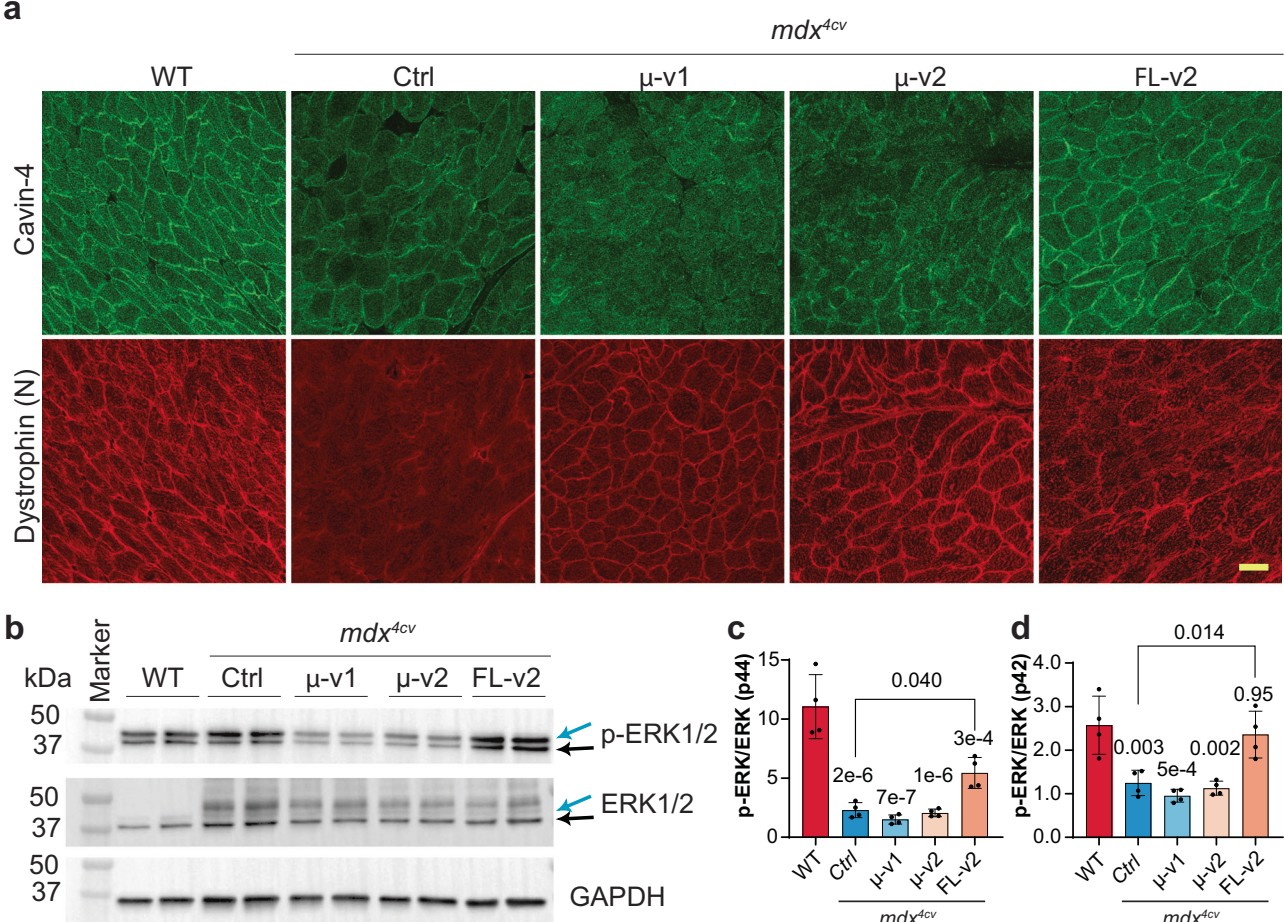

**Fig. 7 | MyoAAV-delivered FL-dystrophin restored the membrane localization of cavin-4 and its associated ERK signaling in the heart muscles of *mdx^4cv* mice.** All animal experiments were performed in WT and *mdx^4cv* male mice with the C57BL/6J genetic background. **a** Immunofluorescence staining of cavin-4 in heart sections of WT and *mdx^4cv* mice treated with or without AAV-FL-v2, AAV-μ-v1, or AAV-μ-v2 (*n* = 3 mice per group). Scale bar: 100 μm. **b**–**d** Western blot and quantification of ERK phosphorylation in the heart muscles of WT and *mdx^4cv* mice treated with or without AAV-FL-v2, AAV-μ-v1, or AAV-μ-v2 (*n* = 4 mice per group). Data were mean ± SEM. Statistical analyses were performed by one-way ANOVA with Tukey's multiple comparisons test except otherwise specified. Source data are provided as a Source Data file.

stabilize the disordered intein[N 48]. Third, stronger enhancer sequences can be tested to drive the N and M expression.

In this proof-of-principle study, we initially utilized a weaker promoter, Spc2-26, to drive the C fragment expression and a stronger Spc5-12 to drive the N and M fragment expression. However, we found that the expression of the C fragment and the resulting FL-dystrophin in the diaphragm muscle was very weak. The weak activity of Spc2-26 likely contributed to the inefficient restoration of FL-dystrophin in the diaphragm. Indeed, when using Spc5-12 to drive the C fragment expression, we observed some improvement in FL-dystrophin expression in the diaphragm and cardiac muscles. Further optimization is required to increase the FL-dystrophin expression in the diaphragm muscle. This could be done by testing different muscle-specific regulatory cassettes, such as those from *DES*, *MYL11*, *SERCA1*, *CKM*, and *FLNC* to drive each fragment expression and their stoichiometry in the AAV cocktail.

Efficient gene replacement therapy for DMD requires a potent AAV capsid with strong muscle tropism. Myotropic capsids have recently been engineered by the directed evolution of AAV capsid libraries containing seven random amino acid insertions into the variable region VIII of AAV9 and in vivo[31,49,50]. These engineered myotropic AAV capsids share a common "RGD" motif, which is known to bind to several integrin heterodimers[51]. In this study, we took advantage of the high muscle tropism of MyoAAV4A, which allowed

simultaneous administration of three AAV vectors at a total dosage that is currently used in clinical trials for neuromuscular diseases. Notably, MyoAAV4A and its related variants were tested in both mouse models and non-human primates[31], highlighting the promise of their clinical translation.

Finally, we performed a comparative study for FL-dystrophin with the triple vector approach vs the micro-dystrophins in *mdx^4cv* mice using the same MyoAAV4A capsid and the same Spc5-12 promoter. Our data showed that delivery of FL-dystrophin and micro-dystrophin at the same total dose (8E + 13 vg/kg) achieved similar levels of functional and histological improvements except for the serum CK measurements. The serum CK levels were normalized to about WT levels in *mdx^4cv* mice treated with micro-dystrophin, but less pronounced in mice treated with FL-dystrophin. The lower dystrophin-positive muscle fibers in diaphragm muscles following FL-dystrophin gene delivery can obviously contribute to the higher CK readings. Interestingly, on WB examination, FL-dystrophin was expressed at ~30–50% of the normal level following the triple vector injection, while micro-dystrophins were expressed at ~2.6–3.7-fold of normal skeletal muscle (Fig. 6b, c). Overall, these data showed that even a lower level of FL-dystrophin expression can provide a similar level of protection in dystrophic skeletal muscle as much overexpressed micro-dystrophins. Moreover, FL-dystrophin expression can restore the membrane localization of cavin-4 and the defective ERK signaling, which could not be

corrected by micro-dystrophin gene delivery in the heart (Fig. 7). We envision that future optimization of the promoter, capsid, and intein splits of triple dystrophin vectors can bring FL-dystrophin expression to bodywide skeletal muscle and heart to offer maximal protection.

One potential concern with FL-dystrophin gene therapy is the host immune response towards the non-self epitopes from FL-dystrophin and its fusion products with inteins, particularly in patients who carry large deletions. A recent study reported that five DMD patients from 4 different clinical trials by Sarepta, Roche (using Sarepta's vector), Pfizer, and Genethon receiving three different gene therapy products differing in AAV serotype, promoter, and dose, showed strikingly similar severe adverse advents that suggested a cytotoxic T-cell immune response against micro-dystrophin proteins[52]. All five patients had similar large overlapping deletions (exon 8 to exon 21), which was present in micro-dystrophins. This highlights the urgent need for specific interventions to prevent immune responses that can limit the efficacy of gene therapy and cause irreparable harm. Such interventions could also be applicable for many other therapeutic approaches under development such as gene editing therapies (in which, the bacteria-derived Cas9 protein is delivered). Although beyond the scope of this study, our future efforts will be concentrated on studying the potential immune responses and the approaches to mitigate them. Encouragingly, a recent study from clinical trials showed that it is possible to prevent antibody response to AAV gene therapy by a cocktail of immune modulators (for example, rituximab plus sirolimus in addition to steroids) to prevent anti-AAV antibody formation[53]. In addition, immune tolerance could be induced by hepatic gene transfer as supported by coagulation factor VIII studies in both animals and a patient with hemophilia A[54]. These types of immune modulation could be tested in the future to see if they can also induce long-term immune tolerance towards the FL-dystrophin gene therapy.

Collectively, our work addresses a key deficiency of current AAV-based gene replacement therapy for DMD by delivering functional FL-dystrophin. The split intein-mediated assembly coupled with potent myotropic AAV capsids enables FL-dystrophin restoration as well as functional, histopathological, and signaling improvement in dystrophic mice. Future studies will be warranted to improve the expression levels and investigate the host immune responses toward FL-dystrophin gene therapy. Moreover, this approach could potentially be implemented for other diseases with large gene payloads exceeding the packaging capacity of AAV to be delivered, such as LAMA2, RYR1, DSP, FLNC, DYSF, OTOF, MYO7A, and F8, among others.

## Methods

### Ethics statement
The C57BL/6J and $mdx^{4cv}$ (B6Ros.Cg-Dmd mdx4cv/J) mice were purchased from the Jackson Laboratory and housed at Indiana University Laboratory Animal Resource Center following animal use guidelines. All the experimental procedures were approved by the Institutional Animal Care and Use Committee of Indiana University (#23003). All mice were maintained under standard conditions of constant temperature ($72 \pm 4$ °F) and humidity (relative, 30–70%), in a specific pathogen-free facility and exposed to a 12/12 h light/dark cycle with ad libitum access to food (standard chow) and water. As DMD primarily affects males, only male mice were used in our experiments.

### Plasmid construction
The plasmid expressing FL-dystrophin (p37-2iDMD-LR) was a gift from Michele Calos (Addgene # 88892). The dystrophin fragments, Cfa, Gp41−1, IMPDH-1, Cat, and VidaL inteins were synthesized at IDT DNA Technologies and amplified by fusion PCR, restriction digested, and ligated into pAAV vectors harboring a mini-CMV promoter with a muscle creatine kinase enhancer (meCMV) and a mini-polyA signal. The plasmids were confirmed by Sanger sequencing and/or whole plasmid sequencing. Detailed information on the plasmids and

primers is provided in Supplementary Table 1 and Supplementary Table 2, respectively.

### Cell culture and transfection
AD293 cells were cultured in Dulbecco's Modified Eagle's Medium (DMEM) (Corning, Manassas, VA) supplemented with 10% fetal bovine serum (FBS) and 1% 100x penicillin-streptomycin solution (10,000 U/ml, Invitrogen). Cells were plated in six-well plates and incubated overnight at 37 °C. When the cultures reached 80% confluence, they were transfected with plasmids expressing FL or dystrophin fragments using PEI (Polysciences Inc., Pennsylvania, USA). At 72 h after transfection, cells were collected for protein extraction.

### AAV vector production, titer determination, and in vivo administration
The MyoAAV4A vectors were produced in suspension 293 cells by triple transfection and purified by OptiPrep Iodixanol gradient ultracentrifugation in the lab or by the Viral Vector Core at Indiana University. Buffer exchange and concentration were carried out using the Amicon centrifugal filter units (MWCO 100 kDa). The final AAV particles were stored in PBS with 0.001% Pluronic F-68 and 200 mM NaCl. All AAV vectors were titered using quantitative real-time PCR (qPCR). Briefly, the AAV particles were treated with Dnase I (Invitrogen, MA, USA) to eliminate any contaminating plasmid DNA, followed by proteinase K (New England Biolabs, MA, USA) digestion to release the transgene. Real-time PCR was performed using PowerUp SYBR Green Master Mix (Applied Biosystems, Thermo Fisher Scientific, MA, USA) in ABI QuantStudio 5 Real-Time PCR System (Applied Biosystems, Thermo Fisher Scientific, MA, USA). Samples were quantified by a standard curve established by serially diluted Sma1-linearized plasmids. Titers are expressed as DNase resistant particles in vector genome per ml (vg/ml): 5.15E + 13 vg/ml for MyoAAV4A-Dys-N1, 4.63E + 13 vg/ml for MyoAAV4A-Dys-M3, 2E + 13 vg/ml for MyoAAV4A-Dys-C6 (Spc2-26), 1.94E + 13 vg/ml for MyoAAV4A-Dys-C6 (Spc5-12), 4.57E + 13 vg/ml for MyoAAV4A-μ-v1, 1.41E + 13 vg/ml for MyoAAV4A-μ-v2. The AAV particles (a total of 2E + 14 or 8E + 13 vg/kg, 2:1:1 of N1, M3 and C6; 8E + 13 vg/kg for MyoAAV4A-μ-v1 or MyoAAV4A-μ-v2) were systemically administered into $mdx^{4cv}$ mice at 3–4 weeks of age through retro-orbital injection as described previously[28].

### Measurement of serum biomarkers
Blood samples were collected at various time points after retro-orbital injection. The blood samples were allowed to clot for 15 min to 30 min and centrifuged at $2300 \times g$ for 10 min at room temperature. The supernatant was collected as serum and stored at −80 °C in small aliquots for the biochemical assays. Measurement of CK (326-10, SEKISUI Diagnostics LLC) was performed according to the manufacturer's instructions.

### Muscle contractility
At 8–12 weeks of age, muscle contractility was measured weekly using an in vivo muscle test system (Aurora Scientific Inc) as described previously[28,41,42]. Mice were anesthetized with 3% (w/v) isoflurane and anesthesia was maintained by 1.5% isoflurane (w/v) during muscle contractility measurement. Maximum plantarflexion tetanic torque was measured during a train of supramaximal electric stimulations of the tibial nerve (pulse frequency 150 Hz, pulse duration 0.2 ms) using the DMA v5.501 (Aurora Scientific Inc).

### Wire hanging assay
The animal was placed on a custom-made wire mesh, then inverted and suspended above a soft cushion. The latency to when the animal falls is recorded. The mouse will be trained two to three times 1 week before the test. This test is performed 3 days per week with two to three trials per session. The average performance for each session is presented as

the average of the trials, and the average for 3 days is used as the average of the mouse.

## Western blot
Cell pellets and mouse tissue samples were lysed using a cold radioimmuno-precipitation analysis buffer (RIPA) supplemented with 1x protease inhibitor cocktail (Thermo Scientific, 78440). Protein concentrations were measured to ensure uniform loading (Bio-Rad DC protein assay kit, 5000111). Proteins were separated using 4–15% pre-cast SDS-PAGE gel (Bio-Rad, 17000927) and transferred onto 0.45 μm nitrocellulose membranes (Bio-Rad, 1620115). After blocking with 5% non-fat dry milk, membranes were incubated with an anti-dystrophin antibody (anti-N-terminus: MANHINGE1B (10F9) or MANHINGE1C (5D12), 1:100, Developmental Studies Hybridoma Bank, Iowa City, IA, USA; anti-M-fragment: MANEX50 (6A9), 1:100, Developmental Studies Hybridoma Bank, Iowa City, IA, USA, or rabbit polyclonal DMD/8773 R, 1:200, NeoBiotechnologies, Union City, CA, USA; anti-C-terminus: ab15277, 1:1000, Abcam, Cambridge, UK), mouse monoclonal anti-p44/42 MAPK (Cell Signaling Technology, 4696, 1:1000), rabbit polyclonal anti-Phospho-p44/42 MAPK (Cell Signaling Technology, 9101, 1:500), or rabbit monoclonal anti-GAPDH antibody (Cell Signaling Technology, 2118 S, 1:2000). Subsequently, membranes were washed and incubated with Horseradish peroxide (HRP)-conjugated goat anti-mouse (7076S, 1:4000, Cell Signaling Technology) and goat anti-rabbit (7074S, 1:4000, Cell Signaling Technology) secondary antibodies. Chemiluminescent detection was employed using enhanced chemiluminescence (ECL) western blotting substrate (Pierce Biotechnology, Rockford, IL, USA), capturing the signal by ChemiDoc XRS+ system (Bio-Rad). Western blots were quantified using ImageJ 1.54f software.

## Histopathological and immunohistochemical assessment of tissues
Mouse tissues (heart, diaphragm, and gastrocnemius) were harvested after euthanasia by $CO_2$, embedded in optimal cutting temperature (OCT, Sakura Finetek, Netherlands) compound, snap-frozen in cold isopentane, and stored at −80 °C. For H&E staining, frozen cryosections (10 μm) of skeletal muscle and heart were fixed in 10% formaldehyde for 5 min at room temperature and then proceeded to the standard protocol of H&E staining. All images were taken under an Axio observer 7 Zeiss microscope (Carl Zeiss Microscopy, LLC, Thornwood, NY, USA). For Masson's trichrome staining, the muscle and heart cryosections were fixed with Bouin's solution for 1 h at 56 °C. After washing with PBS, the tissue sections were stained with Masson's 2000 Trichrome Kit (American MasterTech, Lodi, CA) following the manufacturer's instructions. For immunohistological examinations, frozen cryosections (10 μm) were fixed with 4% paraformaldehyde for 15 min at room temperature. After washing with PBS, the slides were blocked with 3% BSA for 1 h. The slides were incubated with primary antibodies against dystrophin as mentioned above, laminin-α2 (ALX-804–190-C100, 1:100, Enzo Life Sciences Inc, Farmingdale, NY), α-dystroglycan (IIH6 C4, 1:10, Developmental Studies Hybridoma Bank, Iowa City, IA, USA), β-dystroglycan (sc-33702, 1:50, Santa Cruz Biotechnology, Dallas, TX, USA), α-sarcoglycan (ab234589, 1:100, Abcam, Cambridge, UK), β-sarcoglycan (sc-14176, 1:50, Santa Cruz Biotechnology, Dallas, TX, USA), nNOS (sc-5302, 1:50, Santa Cruz Biotechnology, Dallas, TX, USA), α-dystrobrevin (610766, 1:100, Becton Dickinson and Company, NJ, USA), and cavin-4 (ab121647, 1:100, Abcam, Cambridge, UK) at room temperature for 0.5–1 h. The slides were then washed extensively with PBS and incubated with secondary antibodies Alexa Fluor 488 goat anti-rat IgG (A-11006, 1:400, Invitrogen, Carlsbad, CA), Alexa Fluor 594 goat anti-mouse IgG (A11032, 1:400, Invitrogen, Carlsbad, CA), Alexa Fluor Texas Red goat anti-rabbit IgG (T2767, 1:400, Invitrogen, Carlsbad, CA), or Alexa Fluor 568 goat anti-mouse IgM (A21043, 1:400, Invitrogen, Carlsbad, CA) for one hour at room temperature. The slides were sealed with VECTASHIELD Antifade Mounting Medium with DAPI (Vector Laboratory, Burlingame, CA). The images were taken under an Axio observer 7 Zeiss microscope (Carl Zeiss Microscopy, LLC, Thornwood, NY, USA) with Zeiss ZEN version 3.8 (Carl Zeiss Microscopy, LLC, Thornwood, NY, USA) or Leica SP8 Lightning confocal microscope (DMI8, Leica Microsystems, Wetzlar, Germany) equipped with Leica LAS X software (version 1.4.5.27713, Leica Microsystems, Wetzlar, Germany). Laminin-α2-positive and dystrophin-positive muscle fibers, muscle fiber area, and fibrotic area were analyzed using the ImageJ 1.54 f software. The amount of dystrophin-positive muscle fibers is represented as a percentage of total laminin-α2-positive muscle fibers.

## Statistical analysis
The data were expressed as mean ± the standard error of the mean (SEM) and final figures were assembled with Adobe Illustrator 27.8.1. The sample size was estimated using G-power software 3.1. Statistical differences were determined by two-tailed unpaired Student's $t$-test for two groups and one-way ANOVA with Tukey's posttests for multiple group comparisons using GraphPad Prism 10.1.0 (GraphPad Software, La Jolla, CA) with the assumption of Gaussian distribution of residuals. A $p$ value less than 0.05 was considered to be significant.

## Reporting summary
Further information on research design is available in the Nature Portfolio Reporting Summary linked to this article.

## Data availability
All data generated or analyzed during this study are included within the article and its Supplementary Information files. Source data are provided with this paper.

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

## Acknowledgements

The authors thank the Viral Vector Core at Indiana University School of Medicine and Dr. Junping Zhang for providing technical support in AAV production and purification. R.H. is supported by US National Institutes of Health grants (R01HL116546, R01HL159900, R01HL170260, R21HL163720, and R01HL169976).

## Author contributions

R.H. conceived the study and wrote the manuscript. Y.Z. and C.Z. carried out the experiments, analyzed the data, and participated in drafting the manuscript. W.X. and R.W.H. contributed to the final version of the manuscript.

## Competing interests

Indiana University has filed a provisional patent application (inventor: R.H.) based on the work reported in this paper. R.H. is a founder of Zhida Therapeutics. The remaining authors declare no competing interests.
