## [Peer Review File · Nature Communications]

REVIEWER COMMENTS

Reviewer #1 (Remarks to the Author):

As clearly articulated by the authors, current micro-dystrophins in clinical trials provide less than complete rescue of function in patients with DMD so here they have commenced to develop a clever triple vector intein-based method to restore full length dystrophin expression in the mdx mouse model. While the data look promising, they are preliminary in the sense that dystrophin expression relative to WT or healthy, unaffected control muscle was not measured. In addition, no data in the study demonstrate that the full length dystrophin expressed actually provides greater functional restoration than achieved with current micro-dystrophins. Thus, it is left to faith that the full length dystrophin expressed via the triple vector intein method is more effective in mdx mice, or will be in human DMD patients.

While the percentage of dystrophin-positive fibers is high (Fig 3B, C), a similar result has been observed in many gene editing studies where WB data report 10-fold lower expression relative to WT. The current study thus lacks WB quantitation reporting the level of dystrophin expression in the triple vector treated mdx mice compared to WT controls. I appreciate that the human origin of triple vector dystrophin makes it challenging to quantitatively compare with WT murine dystrophin (as is evident from Fig 3D), but the authors should consider quantitating expression with an antibody that reacts more similarly with mouse and human dystrophin, or including unaffected human control muscle lysates for quantitative comparison. As a surrogate/complementary approach, WB of DAPC proteins would provide another quantitative assessment of expression relative to WT/unaffected control muscle that wouldn't suffer from the cross-species issue.

Another limitation of the study is that none of the reported phenotype assessments demonstrate that the (unquantified) level of dystrophin achieved with the triple vector approach actually provides greater (or even the same) functional improvement compared to micro-dystrophins.

Other limitations of the study that should at least be discussed are: the potential of excess Dys-C acting in a dominant negative manner to compromise the function of full-length dystrophin or utrophin (as reported in PMID's 11978768 and 26136477 for example), and considering findings reported in PMID 37314712, the possibility that expression of full length dystrophin in DMD patients with larger sequence deletions might actually fare worse than if they expressed a micro-dystrophin lacking those sequences.

Reviewer #2 (Remarks to the Author):

This is a very elegant paper that describes the development of a triple vector system to deliver full length dystrophin into skeletal and cardiac muscles. The manuscript is well written and the experimental analyses are clear and sound.

My only comment would be to remove the ALT, AST measurements. These are enzymes that are non specific for skeletal muscle and it doesn't add anything significant to the study.

Reviewer #3 (Remarks to the Author):

The manuscript by Zhou, et al. reported the development of a triple-AAV vector system to deliver the full-length (FL) dystrophin, which is one of the largest genes in the human body, into skeletal and cardiac muscles. The authors carefully split the FL dystrophin into three fragments, and engineered split-inteins into the split junctions. Then a series of studies were carried out in vitro to optimize the efficiency of protein trans-splicing and improve the expression of the FL dystrophin in 293 cells. The most efficient three fragments were packaged into AAV vectors by a robust myotropic AAV capsid. After simultaneous administration of three AAV vectors into the dystrophic mouse model, the FL dystrophin was reconstituted in the muscle. Furthermore, the expression of the FL dystrophin improved the histopathology and function of the dystrophic muscle.

Although triple-AAV vectors and split inteins have been exploited in delivering large dystrophin genes in the early studies (Koo, et al. Human Gene Therapy 2014; Lostal, et al. Human Gene Therapy 2014; Li et al. Human Gene Therapy 2008), the reconstitution efficiency in these studies is very low. The most prominent achievement of this manuscript is the significant improvement of the reconstitution efficiency of the FL dystrophin through engineering split-inteins into the split junctions. However, at least in its current form, it remains unclear whether triple-AAV FL dystrophin vectors are better than widely-used AAV micro-dystrophin vectors. Hence, this uncertainty makes it doubtful that this manuscript is of great interest to the broad readership of Nature Communications.

Major Comments:

Currently, AAV micro-dystrophin vectors have been used for DMD gene therapy, as evidenced by the FDA-approved Elevidys. The rationale of developing AAV FL-dystrophin vectors is that FL dystrophin could provide better function than truncated dystrophins. However, given intrinsic limitations of triple-AAV vectors, it is doubtful that this is case.

A. Expression level: Dystrophin expression in the muscle serves as the endpoint in the clinical trial of AAV gene therapy (Boehler, et al. Neuromuscul Disord 2023; Chamberlain, et al. Human Gene Therapy 2023). Do the triple vectors produce the same level of the dystrophin expression as the single vector with the same dose? If not, another concern would be the systemic toxicity associated with higher dose of triple vectors, as seen in the studies of non-human primates and clinical trials;

B. Immune response: Addition of intein motifs introduces extra non-self epitopes, which cause immune response to transgene products, such as by-products of protein splicing, and unassembled fragments. It remains to be determined whether those non-self epitopes from intein engineering causes immune response.

Minor Comments:

1. Fig 1B: On the gel image of Anti-Dystrophin (M), why is the WT control band invisible?

2. Fig 1B: On the gel image of Anti-Dystrophin (N), the band from N -terminal fragment only is barely seen. So the statement outlined from Line 127-129 is weak;

3. Fig 2: More details of how quantification of the band intensity is done are expected. For example, gel images were produced by three different antibodies. Which gel image was included for quantification? For unassembled fragments, which include both two-fragment and one fragment, it is unclear how quantification of unassembled fragments is done.

In summary, this manuscript has demonstrated that reconstitution efficiency of the triple FL dystrophin vectors was significantly improved by split-inteins and a robust myotropic AAV capsid. This technology could be applicable for delivering other large genes. However, the significance of this finding is weakened by ambiguous translational potential and lack of additional important data, such as side-by-side comparison with AAV micro-dystrophin vectors and possible immune response to transgene fragments.

Reviewer #4 (Remarks to the Author):

The authors present triple AAV vector system to deliver full length dystrophin through protein transplicing reaction with use of 2 split inteins. The design of transgene fragments is based on transgene size with junction 1 (Dp260 isoform) and junction 2 (Dp140 isoform). The use of split inteins has been used previously in AAV gene therapies and the authors utilise their previous split intein approach for base -editing for DMD initially using Cfa and Gp41-1 for the 2 junctions between Dys-N, Dys-M and DysC.

Triple transfection of Dys-N, Dys-M and DysC showed weak FL band, strong NM band and strong C band suggesting good assembly with Cfa but not Gp41-1. The authors focussed on DNA ratios 4:2:1 to equalise expression using and iterative processes to optimise the M-C intein assembly switching to IGA-SPT intein. A concern was the significantly higher Dys-C expression compared to N and M, resulting in several iterative processes to reduce Dys-C expression, but eventually found higher level Dys-C improved M-C assembly. To improve assembly and modify Dys-C expression several iterative modifications were made; intron removal, junction site mutation, deleterious change of kozak, reintroduction of strong Kozak, and inclusion of synthetic ubiquitin-dependent proteolysis PB29 to further lower Dys-C expression. The high level of the Dys-C fragment is a potential concern, as this fragment is likely to interact with beta dystroglycan, and hence “depleting” the available pools of beta DG for the full length dystrophin. As an example previous overexpression of Dp71 led to increase pathology. This part is not discussed in the manuscript

Figure 1b: Western blot- There is significant variation in Dys N, M, C in levels of expression when transfected alone. N being very weak, followed by M and C much stronger. This perhaps maybe due to different sensitivities of antibodies for each fragment, but translation efficiency was not assessed for each construct with mRNA transcripts in addition to protein, to evaluate the expression differences between N,M,C constructs.

Given each transgene was expressed by the same meCMV promoter, there was no experimental exploration to optimise/improve Dys N transgene expression. This is surprising, as the level of FL-dystrophin will be limited to the lowest expressing construct (Dys-N) between the 3 fragments, even if the intein assembly is fully optimised.

Figure 2a: the bar charts are labelled blue, red and green, colour-blind palettes are recommended for all charts and schematics if possible.

Figure 2b: protein ladder max size is 250kD. Ideally protein ladders such as HiMark Pre-stained protein ladder showing up to 460kD would be better used to illustrate the differences between FL and NM/ MC bands at 427kD and 290kD respectively. The full length control has a double band at ?250-290 and ?427kD in all the blots.

Typo “Turkey’s multiple comparisons test” throughout manuscript. Should correct to Tukey’s. If there are 3 replicates, it may not be possible to test for Gaussian distribution and non-parametric test should be considered.

In vivo evaluation uses MyoAAV4A with transgenes under transcriptional control of 2 different promoters Spc5-12 for Dys-N and M and Spc2-26 for Dys-C with a 2:1:1: molar ratio. Presumably this is to further equalise expression of the 3 transgenes with different promoter strengths and vector ratio. 13 male mdx4cv mice were treated with IV 2e14vg/kg combined total vg / They demonstrate compelling restoration of dystrophin expression in gastroc and in cardiomyocytes by immuno of 3 separate fragments on different sections. Restoration of DGC and reduction histopathological features, alongside CK and tetanic force and wire hang studies.

Figure 3b: Immuno of Dys -N,M and C are of different sections. It would be more representative to stain same/ consecutive sections with the 3 antibodies and merge to quantify % FL positive fibres, % NM/ MC and % C only.

Figure 3d: Evident that the triple vector system results in a combination of FL, NM, MC and C expression on western blot. The authors should discuss this further, what are the potential contributing functional effects of NM,MC and C expression, given the junction sites were designed to mimic dp260 and 140 isoforms? VGC and mRNA transcripts for each construct could be ascertained to explore if this effect is due to the challenge of transduction with 3 vectors or due to expression efficiency differences between the constructs. Given triple vector systems are also dependent on each cell being transduced by 3 vectors for protein re-assembly, this approach is likely to result in combination of micro, midi and FL dystrophin.

Figure 3e. While the visual assessment is coherent with the interpretation of the authors, semiquantitative digital capture of the immunohistochemistry would allow a much more precise assessment of the restored proteins (a-DB for example appears substantially weaker compared to wild type, but difficult to put a precise number)

The authors should include discussion on alternative approaches to improve the N expression in their constructs. Unfortunately the authors did not compare their approach to microdystrophin benchmark for comparison but attempts to discuss pros and cons of single vs triple vector treatment in terms of translation could be made. While this reviewer does not necessarily think that microdystrophin is an ideal and final candidate for the therapy of patients with DMD, in reality this is the current benchmark.

I notice that CK dropped significantly , but only 3 times after the triple vector transfection, arguably less dramatic drop than the one observed by many authors using microdystrophins.

As also discussed, the ~8% protein expression in diaphragm is disappointing and while it might be related to the vector used, it is not as efficient as what reported by multiple investigators and sponsors involved in DMD research. Because of this, I would suggest not to use terminology such as “dramatic improvement in muscle histopathology and function in a mouse model of DMD”

Finally, are there any immunological considerations in patients with deletions regarding the expression of a full length dystrophin? This should also be discussed

Reviewer #1 (Remarks to the Author):

As clearly articulated by the authors, current micro-dystrophins in clinical trials provide less than complete rescue of function in patients with DMD so here they have commenced to develop a clever triple vector intein-based method to restore full length dystrophin expression in the mdx mouse model. While the data look promising, they are preliminary in the sense that dystrophin expression relative to WT or healthy, unaffected control muscle was not measured. In addition, no data in the study demonstrate that the full length dystrophin expressed actually provides greater functional restoration than achieved with current micro-dystrophins. Thus, it is left to faith that the full length dystrophin expressed via the triple vector intein method is more effective in mdx mice, or will be in human DMD patients.

While the percentage of dystrophin-positive fibers is high (Fig 3B, C), a similar result has been observed in many gene editing studies where WB data report 10-fold lower expression relative to WT. The current study thus lacks WB quantitation reporting the level of dystrophin expression in the triple vector treated mdx mice compared to WT controls. I appreciate that the human origin of triple vector dystrophin makes it challenging to quantitatively compare with WT murine dystrophin (as is evident from Fig 3D), but the authors should consider quantitating expression with an antibody that reacts more similarly with mouse and human dystrophin, or including unaffected human control muscle lysates for quantitative comparison. As a surrogate/complementary approach, WB of DAPC proteins would provide another quantitative assessment of expression relative to WT/unaffected control muscle that wouldn't suffer from the cross-species issue.

Response: We appreciate Reviewer #1 for the thoughtful comments. We have performed new WB to quantify the level of dystrophin expression in the triple vector treated mdx mice using control human skeletal muscle lysate as a reference. Using three different antibodies that recognize N, M and C fragments of human dystrophin, respectively, we showed that the FL-dystrophin was restored to about 32.6% - 49.5% of the normal level in human skeletal muscle. These new data are now provided in Figure 5 and 6.

Another limitation of the study is that none of the reported phenotype assessments demonstrate that the (unquantified) level of dystrophin achieved with the triple vector approach actually provides greater (or even the same) functional improvement compared to micro-dystrophins.

Response: We have performed a new comparative study for the triple vector approach vs two micro-dystrophin constructs in *mdx^{4cv}* mice. Our data showed that: 1) both the triple vector system and the micro-dystrophin constructs increased the force production to about same levels (**Figure 6f**), 2) both the triple vector system and the micro-dystrophin constructs normalized the hanging time of *mdx^{4cv}* mice on the wire hanging assay (**Figure 6g**), 3) both the triple vector system and the micro-dystrophin constructs significantly reduced the serum CK levels, although the micro-dystrophin treatment showed more dramatic reduction (**Figure 6e**), 4) both the triple vector system and the micro-dystrophin constructs significantly improved the histopathology of *mdx^{4cv}* mice (**Figure 6h-j**). Interestingly, on WB examination, the FL-dystrophin was expressed at 32.6% - 49.5% of normal level following the triple vector injection, while micro-dystrophins

were expressed ~2.6-3.7-fold of normal skeletal muscle levels (**Figure 6b, c**). Overall, these data showed that even a lower level of FL-dystrophin expression can provide a similar level of protection in dystrophic skeletal muscle as much overexpressed micro-dystrophins.

Moreover, we provided new data to show that AAV-delivered FL-dystrophin restored the membrane localization of cavin-4 and its associated ERK signaling in *mdx*^{4cv} heart, while micro-dystrophins failed to normalize these defects (**Figure 7**), highlighting the superiority of FL-dystrophin over micro-dystrophin for gene therapy.

Other limitations of the study that should at least be discussed are: the potential of excess Dys-C acting in a dominant negative manner to compromise the function of full-length dystrophin or utrophin (as reported in PMID's 11978768 and 26136477 for example), and considering findings reported in PMID 37314712, the possibility that expression of full length dystrophin in DMD patients with larger sequence deletions might actually fare worse than if they expressed a micro-dystrophin lacking those sequences.

Response: We appreciate reviewer #1 for this great question. PMID 11978768 showed that the Dp260 (MC fragment) restored a stable association between costameric actin and the sarcolemma, assembled the dystrophin-glycoprotein complex, and significantly slowed the progression of muscular dystrophy in *mdx* mice. These results suggest that the MC fragment (as Dp260) would unlikely worsen the disease progression in our triple vector delivery. In PMID 26136477, the authors showed that the NtermDys fragment (corresponding to our NM fragment) did not compete with dystrophin and had no pathological effect. However, PMID 26136477 indeed showed that the 2A-cleaved CtermDys (corresponding to our unassembled C fragment) is sufficient to cause dystrophic cardiomyopathy in transgenic mice when it is overexpressed by over 10 fold relative to full-length dystrophin, and that this high level of CtermDys must be expressed with intact dystrophin <50% of normal levels, to exert dominant-negative peptide-dependent cardiomyopathy. In our study, we showed that the C-fragment was expressed in a level similar to that of full-length dystrophin. Thus, it is unlikely that the unassembled C-fragment will exert a dominant-negative impact. Consistent with this notion, our data did not show increased fibrosis in *mdx* mice treated with the triple vector as observed in PMID 26136477.

In addition, concerns have been raised about the host immune responses towards full length dystrophin in patients with large sequence deletions as recently reported in clinical trials. One potential concern with FL-dystrophin gene therapy is the host immune response towards the non-self epitopes from FL-dystrophin and its fusion products with inteins, particularly in patients who carry large deletions. A recent study reported that five DMD patients from 4 different clinical trials by Sarepta, Roche (using Sarepta's vector), Pfizer and Genethon receiving 3 different gene therapy products differing in AAV serotype, promoter, and dose, showed strikingly similar severe adverse events that suggested a cytotoxic T-cell immune response against micro-dystrophin proteins (PMID: 37314712). All five patients had similar large overlapping deletions (exon 8 to exon 21), which was present in micro-dystrophins. This highlights the urgent need for specific interventions to prevent immune responses that can limit the efficacy of gene therapy and cause irreparable harm. Such interventions could also be applicable for many other therapeutic approaches under development such as gene editing therapies (in which, the bacteria-derived Cas9 protein is delivered). Although beyond the scope of this study, our future

efforts will be concentrated on studying the potential immune responses and the approaches to mitigate them. Encouragingly, a recent study from clinical trials showed that it is possible to prevent antibody response to AAV gene therapy by a cocktail of immune modulators (for example, rituximab plus sirolimus in addition to steroids) to prevent anti-AAV antibody formation and associated immunotoxicities (PMID: 37988172). The Herzog lab has unpublished observations (which will be presented at the 2024 ASGCT meeting) that transient B cell depletion also substantially reduces the risk of CD8+ T cell responses against the transgene product, which is not related to MHC I antigen presentation but may reflect abrogation of T help by CD4+ T cells. Addition of sirolimus further counters T cell responses, which makes such a protocol attractive for immune modulation in gene therapy for DMD. In addition, immune tolerance could be induced by hepatic gene transfer as supported by coagulation factor VIII studies in both animals and a patient with hemophilia A (PMID: 38518767). These types of immune modulation could be tested in the future to see if they can also induce long-term immune tolerance towards the full-length dystrophin gene therapy. We have added this into the Discussion.

Reviewer #2 (Remarks to the Author):

This is a very elegant paper that describes the development of a triple vector system to deliver full length dystrophin into skeletal and cardiac muscles. The manuscript is well written and the experimental analyses are clear and sound.

My only comment would be to remove the ALT, AST measurements. These are enzymes that are non specific for skeletal muscle and it doesn't add anything significant to the study.

Response: We appreciate Reviewer #2 for the endorsement of our work. We have removed the ALT, AST data in the revised manuscript.

Reviewer #3 (Remarks to the Author):

The manuscript by Zhou, et al. reported the development of a triple-AAV vector system to deliver the full-length (FL) dystrophin, which is one of the largest genes in the human body, into skeletal and cardiac muscles. The authors carefully split the FL dystrophin into three fragments, and engineered split-inteins into the split junctions. Then a series of studies were carried out in vitro to optimize the efficiency of protein trans-splicing and improve the expression of the FL dystrophin in 293 cells. The most efficient three fragments were packaged into AAV vectors by a robust myotropic AAV capsid. After simultaneous administration of three AAV vectors into the dystrophic mouse model, the FL dystrophin was reconstituted in the muscle. Furthermore, the expression of the FL dystrophin improved the histopathology and function of the dystrophic muscle.

Although triple-AAV vectors and split inteins have been exploited in delivering large dystrophin genes in the early studies (Koo, et al. Human Gene Therapy 2014; Lostal, et al. Human Gene Therapy 2014; Li et al. Human Gene Therapy 2008), the reconstitution efficiency in these studies is very low. The most prominent achievement of this manuscript is the significant

improvement of the reconstitution efficiency of the FL dystrophin through engineering split-inteins into the split junctions. However, at least in its current form, it remains unclear whether triple-AAV FL dystrophin vectors are better than widely-used AAV micro-dystrophin vectors. Hence, this uncertainty makes it doubtful that this manuscript is of great interest to the broad readership of Nature Communications.

Response: We appreciate Reviewer #3 for the precise summary of our work. We have performed additional studies to compare the FL dystrophin vs two micro-dystrophin vectors. Our results showed that the FL-dystrophin, despite that it was expressed lower than micro-dystrophins, increased the force production and hanging time, and decreased the histopathology and fibrosis to similar levels as the micro-dystrophin vectors (please also see our detailed response above to Reviewer #1's point #2).

Major Comments:

Currently, AAV micro-dystrophin vectors have been used for DMD gene therapy, as evidenced by the FDA-approved Elevidys. The rationale of developing AAV FL-dystrophin vectors is that FL dystrophin could provide better function than truncated dystrophins. However, given intrinsic limitations of triple-AAV vectors, it is doubtful that this is the case.

A. Expression level: Dystrophin expression in the muscle serves as the endpoint in the clinical trial of AAV gene therapy (Boehler, et al. *Neuromuscul Disord* 2023; Chamberlain, et al. *Human Gene Therapy* 2023). Do the triple vectors produce the same level of the dystrophin expression as the single vector with the same dose? If not, another concern would be the systemic toxicity associated with higher dose of triple vectors, as seen in the studies of non-human primates and clinical trials;

Response: First, the new myoAAV allowed us to use a total dose of AAV that does not exceed what has been used in clinical trials. Second, we have performed WB quantification studies, which showed that the FL-dystrophin was restored to about 30-50% of endogenous dystrophin in control human skeletal muscle (please see the new data in Figure 5). We believe that this impressive restoration of full-length dystrophin will be of great interest to the broad readership of Nature Communications and will stimulate broader efforts to further improve the design of the triple vectors or dual vectors to deliver full-length or larger dystrophin to provide maximal protection of dystrophic muscle and heart. Please be mindful that it took over decades of global efforts for testing and optimizing micro-dystrophin gene therapy to reach its clinical approval.

B. Immune response: Addition of intein motifs introduces extra non-self epitopes, which cause immune response to transgene products, such as by-products of protein splicing, and unassembled fragments. It remains to be determined whether those non-self epitopes from intein engineering causes immune response.

Response: We agree that the non-self-epitopes caused by both the full length dystrophin (for patients who has large deletions) and the intein motifs may be of concern for the host immune system. This concern remains valid for many other therapeutic approaches under development such as gene editing therapies (in which, the bacteria-derived Cas9 protein is delivered) or even micro-dystrophin gene therapy itself. Although beyond the scope of this study, our future efforts will be concentrated on studying the potential immune responses and the approaches to mitigate them. Encouragingly, a recent study from clinical trials showed that it is possible to

prevent antibody response to AAV gene therapy by a cocktail of immune modulators (for example, rituximab plus sirolimus in addition to steroids) to prevent anti-AAV antibody formation (PMID: 37988172). As explained in response to Reviewer #1, this protocol also holds potential to prevent CD8+ T cell responses against the transgene product. Alternatives include IL-1R1 blockade combined with TLR9 inhibition (PMID: 38053332), among others. In addition, immune tolerance could be induced by hepatic gene transfer as supported by coagulation factor VIII studies in both animals and a patient with hemophilia A (PMID: 38518767). These types of immune modulation could be tested in the future to see if they can also induce long-term immune tolerance towards the full-length dystrophin gene therapy. We have added this into the Discussion.

Minor Comments:

1. Fig 1B: On the gel image of Anti-Dystrophin (M), why is the WT control band invisible?

Response: The anti-M fragment antibody does not work very well to detect mouse dystrophin, particularly when human dystrophin is overexpressed in cell lysates. It works well for detecting human dystrophin. This is also the reason why we observed opposite expression patterns in Figure 3 when comparing M to N or C antibodies.

2. Fig 1B: On the gel image of Anti-Dystrophin (N), the band from N -terminal fragment only is barely seen. So the statement outlined from Line 127-129 is weak;

Response: Agree. This has been removed.

3. Fig 2: More details of how quantification of the band intensity is done are expected. For example, gel images were produced by three different antibodies. Which gel image was included for quantification? For unassembled fragments, which include both two-fragment and one fragment, it is unclear how quantification of unassembled fragments is done.

Response: Sorry for the confusion. In Figure 2, we used the C antibody blot to quantify the FL/Ctrl; we used the N antibody blot to quantify N/FL and NM/FL; the M antibody blot to quantify M/FL and NM+MC/FL; and the C antibody blot to quantify C/FL and MC/FL. This information is now clearly labeled in the Figure and the text. Similarly, we added such information for the other figures.

In summary, this manuscript has demonstrated that reconstitution efficiency of the triple FL dystrophin vectors was significantly improved by split-inteins and a robust myotropic AAV capsid. This technology could be applicable for delivering other large genes. However, the significance of this finding is weakened by ambiguous translational potential and lack of additional important data, such as side-by-side comparison with AAV micro-dystrophin vectors and possible immune response to transgene fragments.

Response: We have performed a new side-by-side comparison study with AAV micro-dystrophin vectors and the new data is provided in Figure 6, which strengthens the significance of our finding. We hope you will find this work is now acceptable for publication in Nature Communications.

Reviewer #4 (Remarks to the Author):

The authors present triple AAV vector system to deliver full length dystrophin through protein transplicing reaction with use of 2 split inteins. The design of transgene fragments is based on transgene size with junction 1 (Dp260 isoform) and junction 2 (Dp140 isoform). The use of split inteins has been used previously in AAV gene therapies and the authors utilise their previous split intein approach for base -editing for DMD initially using Cfa and Gp41-1 for the 2 junctions between Dys-N, Dys-M and DysC.

Triple transfection of Dys-N, Dys-M and DysC showed weak FL band, strong NM band and strong C band suggesting good assembly with Cfa but not Gp41-1. The authors focussed on DNA ratios 4:2:1 to equalise expression using and iterative processes to optimise the M-C intein assembly switching to IGA-SPT intein. A concern was the significantly higher Dys-C expression compared to N and M, resulting in several iterative processes to reduce Dys-C expression, but eventually found higher level Dys-C improved M-C assembly. To improve assembly and modify Dys-C expression several iterative modifications were made; intron removal, junction site mutation, deleterious change of kozak, reintroduction of strong Kozak, and inclusion of synthetic ubiquitin-dependent proteolysis PB29 to further lower Dys-C expression. The high level of the Dys-C fragment is a potential concern, as this fragment is likely to interact with beta dystroglycan, and hence “depleting” the available pools of beta DG for the full length dystrophin. As an example previous overexpression of Dp71 led to increase pathology. This part is not discussed in the manuscript

Response: We appreciate Reviewer #4 for the clear summary of our work. Please see our response to Reviewer #1, point 3. We have also added this into the discussion (page 17-18).

Figure 1b: Western blot- There is significant variation in Dys N, M, C in levels of expression when transfected alone. N being very weak, followed by M and C much stronger. This perhaps maybe due to different sensitivities of antibodies for each fragment, but translation efficiency was not assessed for each construct with mRNA transcripts in addition to protein, to evaluate the expression differences between N,M,C constructs.

Given each transgene was expressed by the same meCMV promoter, there was no experimental exploration to optimise/improve Dys N transgene expression. This is surprising, as the level of FL-dystrophin will be limited to the lowest expressing construct (Dys-N) between the 3 fragments, even if the intein assembly is fully optimised.

Response: Thank you for this great point. We have begun to optimize the N terminus. Our preliminary data showed that the inteinN is a contributing factor for the instability of the fusion proteins in N and M vectors, and adding a protein stabilization domain can enhance the stability of the N fragment (Fig. a).

Moreover, our preliminary data codon optimization can significantly increase the expression of N fragment,

however, to our surprise, this did not seem to increase the FL-dystrophin assembly (**Fig. b**). We will systemically characterize these new constructs and will report these results in a follow-up manuscript.

Figure 2a: the bar charts are labelled blue, red and green, colour-blind palettes are recommended for all charts and schematics if possible.

Response: Thank you very much for this suggestion. We have changed the graphs to the color-blind palettes.

Figure 2b: protein ladder max size is 250kD. Ideally protein ladders such as HiMark Pre-stained protein ladder showing up to 460kD would be better used to illustrate the differences between FL and NM/ MC bands at 427kD and 290kD respectively. The full length control has a double band at ?250-290 and ?427kD in all the blots.

Response: Thank you. We have performed WB with the HiMark pre-stained protein ladder (see **Figures 5 and 6**).

Typo “Turkey’s multiple comparisons test” throughout manuscript. Should correct to Tukey’s. If there are 3 replicates, it may not be possible to test for Gaussian distribution and non-parametric test should be considered.

Response: Thank you. We have corrected the typo. We have also performed Gaussian distribution and non-parametric test.

In vivo evaluation uses MyoAAV4A with transgenes under transcriptional control of 2 different promoters Spc5-12 for Dys-N and M and Spc2-26 for Dys-C with a 2:1:1 molar ratio. Presumably this is to further equalise expression of the 3 transgenes with different promoter strengths and vector ratio. 13 male mdx4cv mice were treated with IV 2e14vg/kg combined total vg / They demonstrate compelling restoration of dystrophin expression in gastroc and in cardiomyocytes by immuno of 3 separate fragments on different sections. Restoration of DGC and reduction histopathological features, alongside CK and tetanic force and wire hang studies.

Figure 3b: Immuno of Dys -N,M and C are of different sections. It would be more representative to stain same/ consecutive sections with the 3 antibodies and merge to quantify % FL positive fibres, % NM/ MC and % C only.

Response: Thank you very much for this suggestion. We chose a mouse monoclonal anti-N antibody and a rabbit polyclonal anti-M antibody to perform a double staining, similarly, we performed anti-N and anti-C co-staining on consecutive sections. Then analyze the data together. As shown in new **Figure 6** and **Supplementary Figure 7**, we observed that the anti-N, M and C signals almost always co-existed in the same muscle fiber. We can hardly find any muscle fiber that is positive for one but negative for the other.

Figure 3d: Evident that the triple vector system results in a combination of FL, NM, MC and C

expression on western blot. The authors should discuss this further, what are the potential contributing functional effects of NM,MC and C expression, given the junction sites were designed to mimic dp260 and 140 isoforms? VGC and mRNA transcripts for each construct could be ascertained to explore if this effect is due to the challenge of transduction with 3 vectors or due to expression efficiency differences between the constructs. Given triple vector systems are also dependent on each cell being transduced by 3 vectors for protein re-assembly, this approach is likely to result in combination of micro, midi and FL dystrophin.

Response: Our WB can detect only low levels of un-assembled or partially assembled fragments, except the unassembled C fragment. As our co-staining data shown in Figure 3b, we could hardly detect any muscle fiber that is positive for only one or just two fragments. The muscle fibers are either positive for all three antibodies or negative for all three, indicating that the co-transduction rate is high. In cells that were transduced by only one or two of the vectors, such dystrophin fragment proteins are likely unstable and degraded rapidly. Please also see our response to Reviewer #1 regarding the potential impact of the C-fragment.

Figure 3e. While the visual assessment is coherent with the interpretation of the authors, semiquantitative digital capture of the immunohistochemistry would allow a much more precise assessment of the restored proteins (a-DB for example appears substantially weaker compared to wild type, but difficult to put a precise number)

The authors should include discussion on alternative approaches to improve the N expression in their constructs. Unfortunately the authors did not compare their approach to microdystrophin benchmark for comparison but attempts to discuss pros and cons of single vs triple vector treatment in terms of translation could be made. While this reviewer does not necessarily think that microdystrophin is an ideal and final candidate for the therapy of patients with DMD, in reality this is the current benchmark.

Response: We have performed side-by-side comparison with the micro-dystrophin AAV. Please see our response above.

I notice that CK dropped significantly, but only 3 times after the triple vector transfection, arguably less dramatic drop than the one observed by many authors using microdystrophins.

Response: Indeed, we observed less pronounced CK reduction as compared to the use of micro-dystrophins (Figure 6e). This is consistent with the observation that some muscles such as diaphragm had more dystrophin-negative cells following the triple vector delivery as compared to the micro-dystrophin gene delivery. We believe this is because the expression level of FL-dystrophin is much lower than that of micro-dystrophins. We suggest that this could be improved in the future by optimizing the N construct and the promoters utilized.

As also discussed, the ~8% protein expression in diaphragm is disappointing and while it might be related to the vector used, it is not as efficient as what reported by multiple investigators and sponsors involved in DMD research. Because of this, I would suggest not to use terminology such as “dramatic improvement in muscle histopathology and function in a mouse model of DMD”

Response: Thank you. We have changed the word “dramatic” to “substantial”.

Finally, are there any immunological considerations in patients with deletions regarding the expression of a full length dystrophin? This should also be discussed

Response: Please see our detailed response to Reviewer #1 regarding the potential immune concerns.

REVIEWERS' COMMENTS

Reviewer #1 (Remarks to the Author):

The authors have addressed my prior concerns.

Reviewer #4 (Remarks to the Author):

The authors now include 2 additional in vivo efficacy studies with data supporting the potential additional therapeutic scope of FL dystrophin to address cardiac dystrophin deficiency, that is lacking with microdystrophin therapies.

The first additional experiment attempts to address low diaphragmatic FL dystrophin with promoter change for C -Dys to Spc5-12 promoter from Spc2-26 (FLv2). Dose ranging of FLv2 did not result in significant difference and it is then mentioned in text regarding figure 6 the improvement was modest from 8 to 10%.

The Figure 5 western blots Diaphragm panels 8773R and 15277 appear very similar and there is absence of M and C-Dys bands. This does not correspond to the quantification represented in bar charts g,h. This is surprising as a concern of this approach has been the predominant C fragment expression and on these 2 western blots the M and C bands are not evident and appear possibly erroneously duplicated in this figure?

The second additional in vivo study includes comparison with MyoAAV delivered microdystrophins under Spc5-12 promoter. Their FL approach resulted in dystrophin expression at levels of 46.5% of human skeletal muscle in GA. This was significantly lower than microdystrophin expression that were over 2-3 fold higher of human skeletal muscle microdystrophin.

This level showed comparable therapeutic efficacy on behavioural functional studies of muscle contractile force and wire hanging time in all treatment groups. However the CK level for FL was only modest (approx. 3000) whilst microdystrophin appears significantly lower, but statistical analysis between FL groups and microdystrophin groups are not shown (figure 6e). There is significant reduction in CNF and fibrotic area to control mdx4cv mice that is equivocal irrespective of treatment group or dosage.

The authors now highlight a difference following the FL approach to restore cavin 4 localisation and the previously associated ERK signalling. This will not be observed with microdystrophins (unless they express spectrin repeats 14 and 15, as some microdystrophin construct do). No histological differences were observed between FL and microdystrophin groups in cardiac tissue.

Based on previous reviewers comments the authors have addressed the following satisfactorily, reformatting to colourblind palates, correction of typographical errors and running of western blots with approach size ladder for figures 5 and 6 to illustrate proteins at 427kDa. Discussion on the consequences of potential of excess DysC, methods to improve DysN expression and transgene immune responses to FL dystrophin included.

Overall the authors have demonstrated therapeutic application of intein transplicing technology. They now illustrate FL dystrophin restores cavin4 localisation and ERK signalling that may suggest additional therapeutic facet, not delivered by microdystrophin transgenes. However the authors claim in the abstract that this approach is superior to microdystrophins. This should be revised as the authors show comparable clinical efficacy, and while it is correct that the clinical rescue was achieved with lower dystrophin levels, the more significant CK elevation likely indicate a higher level of muscle damage that, for non integrating vecros, will be detrimental. Indeed it is not fully evident from data presented with higher CK, low diaphragmatic expression and the potential risks of C-dys overexpression and transgene immune response that this approach is currently superior to microdystrophin in this preclinical study.